# CURRICULUM GNN-LLM ALIGNMENT FOR TEXT-ATTRIBUTED GRAPHS

## ABSTRACT

Aligning Graph Neural Networks (GNNs) and Large Language Models (LLMs) benefits in leveraging both textual and structural knowledge for Text-attributed Graphs (TAGs) learning, which has attracted an increasing amount of attention in the research community. Most existing literature assumes a uniformly identical level of learning difficulties across texts and structures in TAGs, however, we discover the *text-structure imbalance* problem in real-world TAGs, *i.e.*, nodes exhibit various levels of difficulties when learning different textual and structural information. Existing works ignoring these different difficulties may result in under-optimized GNNs and LLMs with over-reliance on either simplistic text or structure, thus failing to conduct node classifications that involve simultaneously learning complex text and structural information for nodes in TAGs. To address this problem, we propose a novel Curriculum GNN-LLM Alignment (**CurGL**) method, which strategically balances the learning difficulties of textual and structural information on a node-by-node basis to enhance the alignment between GNNs and LLMs. Specifically, we first propose a text-structure difficulty measurer to estimate the learning difficulty of both text and structure in a node-wise manner. Then, we propose a class-based node selection strategy to balance the training process via gradually scheduling more nodes. Finally, we propose the curriculum co-play alignment by iteratively promoting useful information from GNNs and LLMs, to progressively enhance both components with balanced textual and structural information. Extensive experiments on real-world datasets demonstrate that our proposed **CurGL** method is able to outperform state-of-the-art GraphLLM, curriculum learning, as well as GNN baselines. To the best of our knowledge, this is the first study of curriculum alignment on TAGs.

## 1 INTRODUCTION

Text-attributed Graphs (TAGs) are ubiquitous in real-world scenarios (Li et al., 2024b), including academic networks, social networks, website networks, *etc*. These graphs are characterized by nodes with rich textual attributes, providing valuable information for downstream tasks like node classification and link prediction. To leverage both structural and textual information on TAGs, aligning Graph Neural Networks (GNNs) and Large Language Models (LLMs), which excel in structural and textual modeling respectively, has attracted an increasing amount of attention in the research community (Li et al., 2023c; Zhao et al., 2022a; Jin et al., 2023b).

However, we discover the *text-structure imbalance* problem in real-world TAGs, *i.e.*, nodes exhibit various levels of difficulties when learning different textual and structural information, which is neglected in most existing literature. For illustration, we give two examples of varying difficulties across nodes in Figure 1, from the textual and structural perspectives respectively: 1) the learning difficulties of different nodes vary depending on their *textural attributes*. For instance, in the task of predicting the paper (node)'s field (label) on academic networks, some nodes could be simply predicted due to explicit inclusion of *field information* in their texts, while other nodes necessitate a comprehensive understanding of the full text for accurate field prediction; 2) the learning difficulties of different nodes vary depending on their *topological structures* (Wei et al., 2023; Wu et al., 2024). For instance, the nodes near the class boundaries are more challenging to learn compared with those located at the center of their corresponding classes, as boundary nodes interact more frequently with, and are more similar to nodes from other classes.

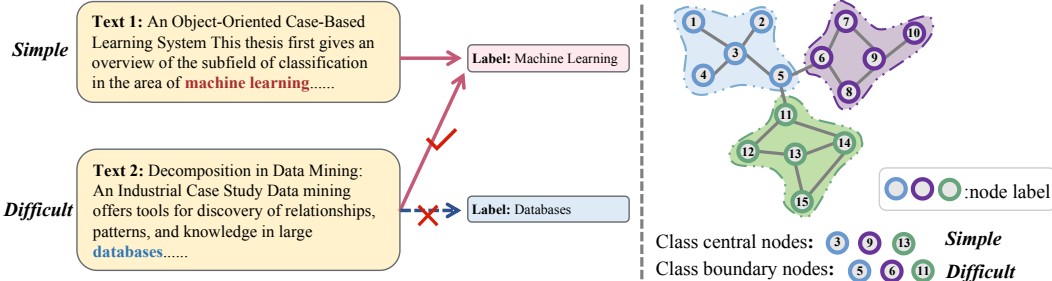

Figure 1: (**Left part**) An illustrative example of the varying difficulty in learning textual attribute semantics from two nodes belongs to Machine Learning in Citeseer Dataset is shown above. Text 1 contains more intuitive semantic information that directly corresponds to the ground truth label, whereas Text 2 requires more complex semantic understanding and even includes potentially misleading information. (**Right part**) An illustrative example of the varying difficulty in learning topological structures is shown above. Class boundary nodes are more challenges than center nodes, due to they interact more frequently with nodes from other classes and are often closer to them.

Existing works ignoring these different difficulties may result in under-optimized GNNs and LLMs, failing to conduct node classifications that involve simultaneously learning complex text and structural information for nodes in TAGs. Specifically, these models may either rely heavily on simplistic textual attributes, overlooking complex yet informative structures for GNNs optimization, or depend mostly on simple structures, ignoring detailed textual content for LLMs optimization. Therefore, the ignorance of different learning difficulties for topologies and texts across nodes inevitably leads to suboptimal performance in TAG tasks that require a deep understanding of both textual and structural information simultaneously.

To address this problem, we propose to balance the text-structure learning difficulty for GNN-LLM alignment in TAGs via employing curriculum learning, which remain unexplored in the existing literature with following key challenges: 1) How to estimate the difficulty of different nodes in TAGs with both structures and texts being taken into consideration; 2) How to make balance in the training process so that more nodes can be scheduled based on their textual and structural difficulties; 3) How to progressively enhance both GNNs and LLMs with balanced textual and structural information during alignment.

To tackle these challenges, we propose a Curriculum GNN-LLM Alignment (**CurGL**) for Text-attributed Graphs. First, we propose a text-structure measurer that accounts for both textual attributes and topological structures to measure the learning difficulty of each node. Particularly, we design a global center-boundary detection method to estimate the topological complexity of structure learning and integrate it with the loss from the LLMs or GNNs to determine the overall text-structure learning difficulty for each node. Second, we propose a class-based node selection strategy that balances the training process by gradually scheduling more nodes. This strategy selects nodes based on their class-specific node's difficulties, while preserving class balance of the selected subgraph. Third, we propose a curriculum co-play alignment to progressively enhance both LLMs and GNNs with balanced textual and structural information. In particular, we iteratively promote useful information obtained from GNNs and LLMs, gradually involving more labeled nodes' and confident pseudo-labeled nodes' textual attributes and topological structures. Extensive experiments on five real-world datasets demonstrate that our proposed **CurGL** method is able to outperform state-of-the-art GraphLLM, curriculum learning, and GNN baselines significantly.

Our main contributions are summarized as follows:

- We propose a novel Curriculum GNN-LLM Alignment (**CurGL**) for TAGs, designed to strategically balance the learning difficulties of textual attributes and topological structures on a node-by-node basis, which is able to significantly improve GNN-LLM alignment and thus enhance the performance. To the best of our knowledge, this is the first study of curriculum alignment on TAGs.

- We propose a difficulty measurer for both textual attributes and topological structures to estimate the learning difficulty in a node-wise manner. Additionally, we introduce a class-based

node selection strategy that selects nodes according to their class-specific node's difficulties while maintaining subgraph class balance.

- We propose a curriculum co-play alignment to iteratively align the LLMs and GNNs by gradually incorporating more labeled and confident pseudo-labeled nodes to promote useful information obtained from both components, thus enhancing them with balanced textual and structural information.

- We conduct extensive experiments on five real-world datasets to demonstrate that our proposed **CurGL** is able to significantly outperform state-of-the-art GraphLLM, curriculum learning, and GNN baselines.

## 2  PROBLEM FORMULATION

In this section, we introduce the fundamental concepts of text-attributed graphs and the notation used in this paper. We focus on the node classification task and describe the learning objectives for LLMs and GNNs on TAGs.

**Text-attributed Graphs.** A text-attributed graph, denoted as $\mathcal{G} = (\mathcal{V}, \mathbf{A}, \mathbf{S})$, consists of a node set $\mathcal{V} = \{v_1, v_2, \ldots, v_N\}$ with $N$ nodes, an adjacency matrix $\mathbf{A} \in \{0, 1\}^{N \times N}$, and a set of textual attributes $\mathbf{S} = \{\mathbf{s}_v\}_{v \in \mathcal{V}}$, where $\mathbf{s}_v$ is the textual attribute of node $v$.

**Node Classification Task.** The node classification task aims to predict the labels of unlabeled nodes within the same label space as the labeled nodes in a given graph. Formally, given a graph $\mathcal{G} = (\mathcal{V}, \mathbf{A}, \mathbf{S})$ with a label set $\mathcal{Y}$, the task is to predict the labels of the unlabeled nodes $\mathcal{V}_U = \mathcal{V} \setminus \mathcal{V}_L$, where $\mathcal{V}_L$ is the set of labeled nodes. The objective is to learn a function $f : (\mathbf{A}, \mathbf{S}) \to \mathcal{Y}$ that maps the node embeddings to the label space.

**Large Language Models for Node Classification.** The node classification task on TAGs for LLMs can be framed as a text classification task (Socher et al., 2013). LLMs aim to leverage the sentence $\mathbf{s}_v$ associated with each node $v$ for label prediction. For a LLM $f$ with parameters $\theta$ for node classification task, the prediction process can be described as follows:

$$\hat{\mathbf{y}}_v = \text{Softmax}(\text{MLP}(\mathbf{h}_v)), \tag{1}$$
$$s.t. \ \mathbf{h}_v = \text{SeqEnc}(\mathbf{s}_v),$$

where $\text{SeqEnc}(\cdot)$ is a text encoder that projects the sentence $\mathbf{s}_v$ into a vector representation $\mathbf{h}_v$, and $\hat{\mathbf{y}}_v$ represents the predicted logits for node $v$. The training objective of the LLM $f_\theta$ can be defined as follows:

$$\min_\theta \mathbb{E}_{p(\mathbf{y}, \mathbf{S})} \mathcal{L}(f_\theta(\mathbf{S}), \mathbf{y}) \tag{2}$$

**Graph Neural Networks for Node Classification.** The node classification task on TAGs for GNNs requires an initial numerical representation for each node, which can be derived using text encoding methods such as bag-of-words (Harris, 1954), TF-IDF (Salton & Buckley, 1988), or Pretrained Language Models (PLMs). GNNs then employ a message-passing mechanism to iteratively update these node representations. For a GNN $g$ with parameters $\phi$ for node classification task, the prediction process can be described as follows:

$$\hat{\mathbf{y}}_v = \text{Softmax}(\mathbf{h}_v^L), \tag{3}$$
$$s.t. \ \mathbf{h}_v^l = \text{COM}^l(\mathbf{h}_v^{l-1}, \text{AGG}(\mathbf{h}_u^{l-1} : u \in \mathcal{N}(v))),$$

where $\text{COM}(\cdot)$ and $\text{AGG}(\cdot)$ represent the combination and aggregation functions, respectively, $\mathbf{h}_v^l$ denotes the node representation of node $v$ at layer $l$, and $\mathcal{N}(v)$ denotes the neighbor nodes of node $v$. The training objective of the GNN $g_\phi$ can be defined as follows:

$$\min_\phi \mathbb{E}_{q(\mathbf{y}, \mathcal{G})} \mathcal{L}(g_\phi(\mathcal{G}), \mathbf{y}). \tag{4}$$

## 3  METHOD

In this section, we introduce Curriculum GNN-LLM Alignment (**CurGL**), a method designed to strategically balance the learning difficulties of textual attributes and structures on a node-by-node

basis. By progressively learning both the textual attributes and topological structures of nodes, our approach enhances the alignment between GNNs and LLMs. The framework of our method is illustrated in Figure 2.

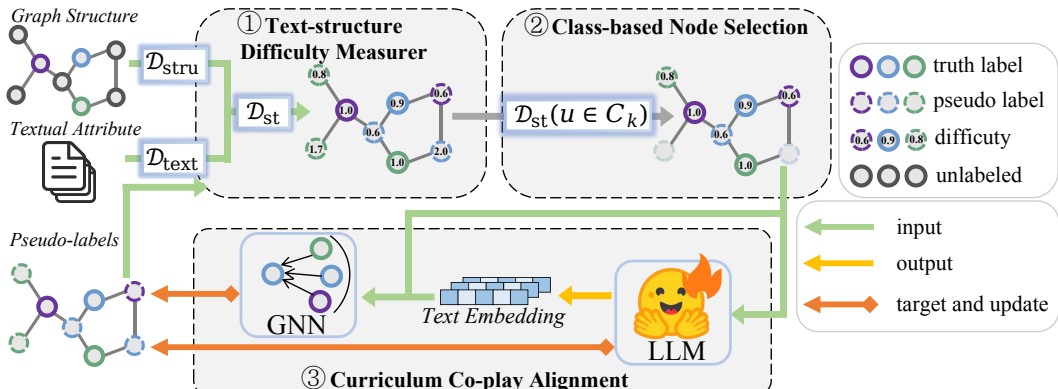

Figure 2: The framework of our proposed method (**CurGL**). The framework consists of three main modules: text-structure difficulty measurer, class-based node selection, and curriculum co-play alignment. (**Part ①**) We calculate the learning difficulty of both text and structure for each node using the text-structure difficulty measurer. (**Part ②**) Then candidate nodes are selected based on their class-specific difficulty levels through a class-based node selection strategy. (**Part ③**) We iteratively align the LLM and GNN by gradually involve more the labeled and confident pseudo-labeled nodes to balance the learning of textual attributes and topological structures in TAGs.

### 3.1 TEXT-STRUCTURE DIFFICULTY MEASURER

Previous curriculum graph learning methods primarily focused on designing difficulty measurers for nodes or edges by considering the graph structure, such as class diversity among a node's neighbors (Wei et al., 2023). However, these methods lack a global structure difficulty measurement and overlook difficulty of the textual attributes of the nodes, limiting their applicability to TAGs. To address this, we propose a text-structure difficulty measurer that accounts for both textual attributes and topological structures to estimate the learning difficulty of each node. Specifically, we design a global center-boundary detection to measure the difficulty of learning the topological structure from a global view, and combine this with the loss of the LLM or GNN to estimate the overall difficulty of learning both the textual attributes and topological structures of each node.

**Structure Difficulty.** For structure difficulty, we introduce a global center-boundary detection method to measure the difficulty of learning the topological structure of each node. This method assumes that nodes located near class boundaries are more challenging to learn, while those at the center of a class should be prioritized for earlier learning. The global center-boundary detection method is defined as follows:

$$D_s(i) = \frac{1}{N(N-1)} \Big( \sum_{\substack{u \neq i \neq v \\ \mathbf{y}_u \neq \mathbf{y}_v}} \frac{\sigma_{u,v}(i)}{\sigma_{u,v}} - \gamma \sum_{\substack{u \neq i \neq v \\ \mathbf{y}_u = \mathbf{y}_i = \mathbf{y}_v}} \frac{\sigma_{u,v}(i)}{\sigma_{u,v}} \Big), \tag{5}$$

where $\sigma_{u,v}$ denotes the number of shortest paths from $u$ to $v$, and $\sigma_{u,v}(i)$ denotes the number of shortest paths from $u$ to $v$ that pass through $i$.

The first term in the equation is designed to detect the boundary degree of a node. Consider a node $i$ and a pair of nodes $u$ and $v$, which belong to different classes. The denominator $\sigma_{u,v}$ denotes the number of shortest paths from $u$ to $v$, while the numerator $\sigma_{u,v}(i)$ denotes the number of shortest paths from $u$ to $v$ that pass through $i$. The larger the first term, the more likely node $i$ is to be positioned at the boundary between two classes of $u$ and $v$, indicating that node $i$ is harder to learn. The second term in the equation aims to detect the center degree of a node. Consider again a $i$ and a pair of nodes $u$ and $v$, but the nodes $i$, $u$ and $v$ belong to the same class. As before, the denominator

$\sigma_{u,v}$ represents the number of shortest paths from $u$ to $v$, and the numerator $\sigma_{u,v}(i)$ represents the number of shortest paths from $u$ to $v$ through $i$. The larger the second term, the more likely node $i$ is to be positioned at the center of its class, indicating that node $i$ is easier to learn and should be prioritized for earlier learning.

Our approach is inspired by Class-Conditional Betweenness Centrality (Wu et al., 2024), and we extend this method to measure the difficulty of learning the topological structure of each node in TAGs. We discuss the differences and provide an intuitive example in Appendix A.1 to demonstrate how the method works.

**Text Difficulty.** For text difficulty, we can just use the loss of LLM to measure the text semantic difficulty when selecting node for GNN training, as LLMs are specifically designed to capture textual semantics. However, we can also use the loss of GNN to measure the text difficulty when selecting node for LLM tuning, which can help to balance the learning of text and structure. Specifically, we define the whole difficulty measurer as follows:

$$D_{st}(v) = D_s(v) + \beta D_t(v) \quad s.t. \ D_t(v) = 1 - \hat{\mathbf{y}}_v, \tag{6}$$

where $\mathbf{y}_v$ is the label of node $v$, and $\hat{\mathbf{y}}_v$ represents the output logits of node $v$ from LLM or GNN. The hyperparameter $\beta$ controls the balance between text and structure difficulty.

Note that the difficulty measurer varies depending on whether we are tuning the LLM or training the GNN. For LLM tuning, we utilize the output logits $\hat{\mathbf{y}}$ from the GNN, whereas for GNN training, we rely on the output logits $\hat{\mathbf{y}}_i$ from the LLM. The value of $\beta$ is adjusted accordingly: during LLM tuning, where the focus is on text learning, $\beta$ is set to a larger value; conversely, during GNN training, where structure learning is prioritized, $\beta$ is set to a smaller value.

---

**Algorithm 1** Class-based Node Selection

---

**Input:** Current training step $t$, total number of training steps $T$, current candidate nodes $\mathcal{V}_C^t$, node difficulty $D_{st}$, and initial proportion of nodes selected from each class $\lambda_0$.
**Output:** Selected nodes $\mathcal{V}_S^t$.
1: Initialize $\mathcal{V}_S^t = \emptyset$
2: Calculate $\lambda_t$ using Eq. 8
3: Devide $\mathcal{V}_C^t$ into $K$ classes, $\mathcal{V}_C^t = [\mathcal{V}_{C_1}^t, \mathcal{V}_{C_2}^t, \dots, \mathcal{V}_{C_K}^t]$
4: **for** $k = 1$ to $K$ **do**
5:     Sort $\mathcal{V}_k^t$ based on $D_{st}$ in ascending order
6:     Select $\lambda_t$ of nodes from $\mathcal{V}_k^t$ and add them to $\mathcal{V}_S^t$
7: **end for**

---

## 3.2 CLASS-BASED NODE SELECTION

To balance the learning difficulties of textual attributes and structures in TAGs, we need to gradually involve more nodes in the training process, considering the varying difficulty levels across different nodes. However, involving nodes solely based on their difficulty $D_{st}$ may overlook the fact that the difficulty of nodes varies across different classes, and potentially leading to an imbalanced class distribution within the subgraph.

To consider the varying difficulty levels across different nodes and classes, we propose a class-based node selection strategy to balance the training process by gradually scheduling more nodes. This approach selects nodes based on their class-specific node's difficulty levels while maintaining sampled subgraph class balance. Specifically, at each step $t$, we divide the candidate nodes $\mathcal{V}_C^t$ into $K$ classes, denoted as $\mathcal{V}_C^t = [\mathcal{V}_{C_1}^t, \mathcal{V}_{C_2}^t, \dots, \mathcal{V}_{C_K}^t]$, where $\mathcal{V}_{C_k}$ represents the candidate nodes in class $k$. We then select low-difficulty nodes $\mathcal{V}_S^t$ from each class in a proportion to $\lambda_t$, which is controlled by a PacingFunction($\cdot$):

$$\mathcal{V}_S^t = \bigcup_{k=1}^{K} \text{Sample}(\lambda_t, \mathcal{V}_{C_k}^t, D_{st}) \quad s.t. \ \mathcal{V}_{C_k}^t = \{v_i | v_i \in \mathcal{V}_C^t \wedge v_i \in \mathcal{V}_k\} \tag{7}$$

$$\lambda = \min(1, \ \text{PacingFunction}(\lambda_0, t)) \quad s.t. \ \lambda_0 \in [0, 1], t \in [1, T], \tag{8}$$

where $\mathrm{Sample}(\cdot)$ denotes the sampling function that selects $\lambda_t$ proportion of nodes from $\mathcal{V}_{C_k}^t$ based on the difficulty $D_{st}$ of each node in $\mathcal{V}_{C_k}^t$, $\mathcal{V}_k$ represents the nodes in class $k$, $\lambda_0$ is the initial proportion of candidate nodes selected, $T$ is the total number of training steps, and $t \in \{1, \ldots, T\}$ is the current training step. The Pacing Function is a monotonically increasing function of the training step $t$, controlling the proportion of nodes $\lambda_t$ selected from each class at each step. The algorithm is provided in the Algorithm 1.

---

**Algorithm 2** Curriculum GNN-LLM Alignment For TAGs

---

**Require:** Total number of training steps $T$, initial proportion of nodes selection $\lambda_0$, confidence pseudo-label hyperparameter $\alpha_l$ and $\alpha_g$, difficulty balance hyperparameter $\beta_l$ and $\beta_g$, labeled nodes $\mathcal{V}_L$, and unlabeled nodes $\mathcal{V}_U$.
1: Pretrain LLM $f^p$ and GNN $g^p$ on $\mathcal{V}_L$ to obtain initial pseudo-labels $\hat{\mathbf{y}}$ of $\mathcal{V}_U$
2: **for** $t = 1$ to $T$ **do**

  3:   **LLM E-step:** $\mathcal{V}_S^t = \emptyset$           9:   **GNN M-step:** $\mathcal{V}_S^t = \emptyset$
  4:   Obtain $\mathcal{V}_C^t$ using Algorithm 3       10:  Obtain $\mathcal{V}_C^t$ using Algorithm 3
  5:   Calculate $D_{st}$ in $\mathcal{V}_C^t$ using Eq. 6     11:  Calculate $D_{st}$ in $\mathcal{V}_C^t$ using Eq. 6
  6:   Obtain $\mathcal{V}_S^t$ using Algorithm 1        12:  Obtain $\mathcal{V}_S^t$ using Algorithm 1
  7:   Train LLM $f_\theta$ as in Eq. 9           13:  Train GNN $g_\phi$ as in Eq. 10
  8:   Update Pseudo-labels $\hat{\mathbf{y}}$           14:  Update Pseudo-labels $\hat{\mathbf{y}}$

15: **end for**

---

### 3.3 Curriculum Co-play Alignment

To align the LLM and GNN in learning both complex textual attributes and topological structures of nodes in TAGs, we iteratively promote useful information from LLM and GNN, progressively enhancing both components with a balanced textual and structural information, *i.e.*, ensuring that the learning capabilities of the current LLM and GNN respectively correspond to the textual and structural complexities of the nodes. Specifically, we propose a curriculum co-play alignment that gradually incorporates more complex textual attributes and topological structures of nodes, iteratively improving the learning of both text and structure via an easy-to-hard learning strategy. The GNN primarily focuses on the progressively increasing topological complexity, while the LLM emphasizes the growing difficulty of textual information. Therefore, the parameter $\beta$ is set to a larger value during LLM tuning and a smaller value during GNN training, as described in Section 3.1.

By employing a curriculum strategy, we prioritize simpler nodes at the beginning of the alignment process. These simpler unlabeled nodes are more likely to be accurately predicted by the LLM and GNN, making their pseudo-labels more reliable and suitable for training. Consequently, we add a portion of the pseudo-labeled nodes to the candidate node set $\mathcal{V}_C$ for further selection and training. This approach offers two benefits: 1) it encourages the GNN and LLM to mutually learn the textual and structural information of these unlabeled nodes; 2) adding the pseudo-labeled nodes to the candidate set $\mathcal{V}_C$ enables a more accurate calculation of the topological complexity of the nodes, as the sampled subgraph from $\mathcal{V}_C$ becomes more representative of the entire graph. We adopt the output logits $\hat{\mathbf{y}}$ as the pseudo-label for each unlabeled node $v$ in $\mathcal{V}_U$.

Specifically, we initially select $\alpha$ proportion nodes from the unlabeled set $\mathcal{V}_U$ based on the output logits $\hat{\mathbf{y}}$ of the LLM and GNN, adding them to the pool of candidate nodes $\mathcal{V}_C$ for further selection. Besides, to ensure the class balance of the subgraph, we also adopt a class-based strategy to select the unlabeled nodes from each class. The detailed algorithm to obtain the candidate nodes set $\mathcal{V}_C$ from the labeled nodes $\mathcal{V}_L$ and the unlabeled nodes $\mathcal{V}_U$ is provided in the Algorithm 3 in the Appendix A.4. We then select nodes from the candidate nodes set $\mathcal{V}_C$ based on the text-structure difficulty $D_{st}$ of each node for training, as described in Section 3.2. For edges, we retain the edges connecting these selected nodes $\mathcal{V}_S$ to form a subgraph. Inspired by (Zhao et al., 2022a), we adopt a variational EM algorithm to iteratively align the LLM and GNN through Curriculum Text-Structure Learning. In the E-step, the GNN is fixed, and the LLM uses only the text information to predict the labels (including both the labeled nodes and the pseudo-labeled nodes inferred by the GNN). In the M-step, the LLM is fixed, and the GNN uses the text embeddings from LLM to predict the labels.

Furthermore, pseudo-labels are continuously updated as the process progresses, with more confident pseudo-labeled nodes being labeled.

We first pretrain an additional LLM model, $f^p$, and a GNN model, $g^p$ (the LLM model and GNN model that are subsequently trained are denoted as $f_\theta$ and $g_\phi$, respectively). Specifically, we train the LLM model $f^p$ on the labeled node set $\mathcal{V}_L$, using $\mathbf{s}_N$ as the input and $\mathbf{y}_L$ as the target, to obtain the text embeddings $\mathbf{h}_N$. Subsequently, we pretrain the GNN model $g^p$ on $\mathcal{V}_L$, using $\mathbf{h}_N$ as the input and $\mathbf{y}_L$ as the target, to generate initial pseudo-labels $\hat{\mathbf{y}}$ for the unlabeled node set $\mathcal{V}_U$. We then iteratively optimize the LLM $f$ with parameters $\theta$ and the GNN $g$ with parameters $\phi$ using the variational EM algorithm.

**LLM Optimization (E-step).** We then train the LLM model $f$ with parameters $\theta$ on the selected node set $\mathcal{V}_S$ with $\mathbf{s}_N$ as input and $\mathbf{y}_L, \hat{\mathbf{y}}_U$ as the target. We denote distribution $q$ as the LLM model and $p$ as the GNN model. The training objective is defined as follows:

$$\min_\theta \sum_{v \in L} \mathbb{E}_{p(\mathbf{y}_v, \mathbf{s}_v)} \mathcal{L}(f_\theta(\mathbf{s}_v), \mathbf{y}_v) + \sum_{v \in U} \mathbb{E}_{p(\hat{\mathbf{y}}_v, \mathbf{s}_v)} \mathcal{L}(f_\theta(\mathbf{s}_v), \hat{\mathbf{y}}_v) \qquad (9)$$

where $\mathcal{L}$ is the Cross-Entropy loss function, $\mathbf{y}_v$ is the ground-truth label of node $v$, and $\hat{\mathbf{y}}_v$ is the output logits $v$ from GNN. And then we update text embeddings $\mathbf{h}_N$ and the pseudo-labeled nodes $\mathcal{V}_U$ based on the output logits $\hat{\mathbf{y}}$ of the LLM model $f_\theta$.

**GNN Optimization (M-step).** We train the GNN model $g_\phi$ on the selected node set $\mathcal{V}_S$ with $\mathbf{h}_N$ as input and $\mathbf{y}_L, \hat{\mathbf{y}}_U$ as the target. The training objective is defined as follows:

$$\min_\phi \sum_{v \in L} \mathbb{E}_{q(\mathbf{y}_v, \mathcal{G}_v)} \mathcal{L}(g_\phi(\mathcal{G}_v), \mathbf{y}_v) + \sum_{v \in U} \mathbb{E}_{q(\hat{\mathbf{y}}_v, \mathcal{G}_v)} \mathcal{L}(g_\phi(\mathcal{G}_v), \hat{\mathbf{y}}_v), \qquad (10)$$

where $\mathcal{G}_v$ is the ego-graph of node $v$ and $\hat{\mathbf{y}}_v$ is the output logits $v$ from LLM. And then we update the pseudo-labeled nodes $\mathcal{V}_U$ based on the output logits $\hat{\mathbf{y}}$ of the GNN model $g_\phi$. The overall algorithm is provided in the Algorithm 2.

## 4 EXPERIMENT

In this section, we conduct extensive experiments to veriy that our framework outperforms existing GraphLLM approaches in TAGs learning tasks.

### 4.1 REAL-WORLD DATASETS

**Baseline.** We adopt several representative GNNs, Curriculum Graph Learning, Pretrained Language Models, and GraphLLM methods as baselines to compare with our approach on real-world datasets. We categorize the GraphLLM methods into three groups based on their training strategies (Jin et al., 2023a; Li et al., 2023c). A more detailed description of these baselines is provided in Appendix C.1.

**Datatsets.** We evaluate our method on five real-world datasets: Cora, Citeseer, Pubmed, Ogbn-arxiv, and WikiCS. The statistics of the datasets and train-validation-test splits are summarized in Table 2. These datasets are widely used in the literature including citation networks and web link networks ands contain nodes with textual attributes and labels. We provide a more description of each dataset in Appendix C.2.

**Exprimental Setting.** We follow the same experimental setup as in GraphLLM benchmark (Li et al., 2024b), utilizing the same train-validation-test split provided by the original datasets. The evaluation metrics used are consistent with prior studies: we report both Accuracy and Macro-F1 scores for the node classification task. To ensure a fair comparison, we employ Sentence-BERT as the LLM model and GraphSAGE as the GNN model, in line with the settings used in the GraphLLM baseline.

**Exprimental Results.** Based on the results in Table 1, we make the following observations: (1) GraphLLM methods achieve better performance than GNN-only methods, demonstrating the effectiveness of leveraging LLMs for learning node text representations in TAGs. (2) Pretrained Language Models methods do not surpass GNNs methods or the GraphLLM approach, indicating that graph structure information is essential for learning node representations in TAGs. (3) Our method outperforms all baselines across all datasets. This superiority can be attributed to our method's ability

Table 1: Accuracy and Macro-F1 results (%) of different methods on real-world datasets. The highest result is **bold**, while the second-best result is marked with underline.

| Model | Cora | | Citeseer | | Pubmed | | Ogbn-arxiv | | WikiCS | |
|---|---|---|---|---|---|---|---|---|---|---|
| | Acc | F1 | Acc | F1 | Acc | F1 | Acc | F1 | Acc | F1 |
| GCN | 82.11 | 80.65 | 69.84 | 65.49 | 79.10 | 79.19 | 72.24 | 51.22 | 80.35 | 77.63 |
| GAT | 80.31 | 79.00 | 68.78 | 62.37 | 76.93 | 76.75 | 71.85 | 52.38 | 79.73 | 77.40 |
| GraphSAGE | 79.88 | 79.35 | 68.23 | 63.10 | 76.79 | 76.91 | 71.88 | 52.14 | 79.87 | 77.05 |
| CLNode | 82.79 | 81.83 | 67.23 | 62.90 | 79.22 | 79.23 | 46.94 | 15.34 | 78.95 | 74.93 |
| RCL | 76.74 | 75.46 | 63.79 | 60.15 | 79.98 | 80.26 | 54.31 | 30.35 | 78.63 | 75.72 |
| TSS | 82.05 | 80.70 | 66.03 | 61.59 | 77.71 | 77.71 | OOM | OOM | 79.04 | 75.50 |
| Sent-BERT (22M) | 69.73 | 67.59 | 68.39 | 64.97 | 65.93 | 67.33 | 72.82 | 53.43 | 77.07 | 75.11 |
| BERT (110M) | 69.71 | 67.53 | 67.77 | 64.10 | 63.69 | 64.93 | 72.29 | 53.30 | 78.55 | 75.74 |
| RoBERTa (355M) | 69.68 | 67.33 | 68.19 | 64.90 | 71.25 | 72.19 | 72.94 | 52.70 | 78.67 | 76.16 |
| GIANT | 81.04 | 80.13 | 65.82 | 62.31 | 76.89 | 76.05 | 72.04 | 50.81 | 80.48 | 78.67 |
| TAPE | 80.95 | 79.79 | 66.06 | 61.84 | 79.87 | 79.30 | 72.99 | 51.43 | 82.33 | 80.49 |
| OFA | 75.24 | 74.20 | 73.04 | 68.98 | 75.61 | 75.60 | 73.23 | 57.38 | 77.34 | 74.97 |
| ENGINE | 81.54 | 79.82 | 72.15 | 67.65 | 74.74 | 75.21 | 75.01 | 57.55 | 81.19 | 79.08 |
| InstructGLM | 69.10 | 65.74 | 51.87 | 50.65 | 71.26 | 71.81 | 39.09 | 24.65 | 45.73 | 42.70 |
| GraphText | 76.21 | 74.51 | 59.43 | 56.43 | 74.64 | 75.11 | 49.47 | 24.76 | 67.35 | 64.55 |
| GraphAdapter | 72.85 | 70.66 | 69.57 | 66.21 | 72.75 | 73.19 | 74.45 | 56.04 | 70.85 | 66.49 |
| LLaGA | 74.42 | 72.50 | 55.73 | 54.83 | 52.46 | 68.82 | 72.78 | 53.86 | 73.88 | 70.90 |
| GLEM$_{GNN}$ | 82.11 | 80.00 | 71.16 | 67.62 | 81.72 | 81.48 | 76.43 | 58.07 | 82.40 | 80.54 |
| GLEM$_{LLM}$ | 73.79 | 72.00 | 68.78 | 65.32 | 79.18 | 79.25 | 74.03 | 58.01 | 80.23 | 78.30 |
| Patton | 70.50 | 67.97 | 63.60 | 61.12 | 84.28 | 83.22 | 70.74 | 49.69 | 80.81 | 77.72 |
| **CurGL** | **85.49** | **84.00** | **73.92** | **69.75** | **85.07** | **84.56** | **76.77** | **60.74** | **82.43** | 80.79 |

to progressively align GNN and LLM, balancing the learning of textual attributes and topological structures in TAGs, resulting in improved node representation learning. Additionally, the higher Macro-F1 scores achieved by our method indicate balanced performance across all classes, which can be attributed to the class-based node selection strategy.

## 4.2 ABLATION STUDY

In this section, we conduct ablation studies to verify the effectiveness of the key modules in our method.

**Pseudo-label.** We first remove the pseudo-label and use only the labeled nodes as the candidate nodes for training. As shown in Figure 3, the performance of our method drops significantly without the pseudo-label selection strategy, indicating that learning the textual attributes and topological structures of confident pseudo-labeled nodes enhances model performance.

**Class-based node selection strategy.** We further remove the class-based node selection strategy described in Section 3.2 and select the nodes solely based on their difficulty. The results in Figure 3 show that the absence of this strategy reduces the performance of our method, demonstrating that selecting nodes based on difficulty while maintaining class balance in the subgraph results in a more balanced performance across all classes.

**Text-structure difficulty measurer.** We further remove the text-structure difficulty measurer described in Section 3.1 and select nodes solely based on the loss. The

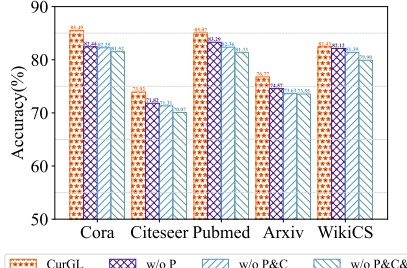

Figure 3: Ablation studies of (**CurGL**), where 'w/o P' denote remove the pseudo-label, 'w/o P&C' denote further remove the class-based node selection strategy, and 'w/o P&C&D' denote further remove the text-structure difficulty measurer.

results in Figure 3 reveal a decline in performance without the text-structure difficulty measurer, highlighting its effectiveness in measuring the learning difficulty of each node. Additionally, training the model with a curriculum strategy further improves performance in TAGs learning tasks.

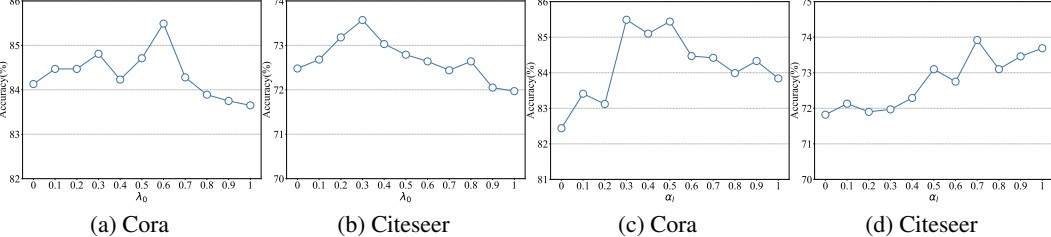

|     |     |     |     |
| --- | --- | --- | --- |
| (a) Cora | (b) Citeseer | (c) Cora | (d) Citeseer |

Figure 4: **(a)(b)**The test accuracy of **CurGL** on the test set as increasing $\lambda_0$ on Cora and Citeseer. **(c)(d)** The test accuracy of **CurGL** on the test set as increasing $\alpha_l$ on Cora and Citeseer.

### 4.3 HYPERPARAMETER ANALYSIS

In this section, we conduct a hyperparameter analysis to examine the impact of different hyperparameters on our method. We evaluate the performance of our method under various hyperparameter settings on real-world datasets. The hyperparameter $\lambda_0$ in Eq.8 represents the initial selection proportion of candidate nodes, and we adjust its the value from 0 to 1. The results in Figure 4 (a) and (b) show that $\lambda_0$ is crucial for the performance of our method. When $\lambda_0$ is set to 1, meaning that all candidate nodes are selected for initial training, the performance drops significantly, indicating that the curriculum learning strategy is essential for the success of our method.

The hyperparameter $\alpha_l$ in Algorithm 3 controls the number of selected pseudo-labeled nodes, and we adjust its value between 0 and 1. As shown in Figure 4 (c) and (d), $\alpha_l$ plays a crucial role in performance. When $\alpha_l$ is set to 0, meaning no pseudo-labels are selected, the performance of our method drops significantly, highlighting the importance of the curriculum confident pseudo-label alignment strategy. However, setting $\alpha_l$ too high can degrade performance, likely due to the increased noise in the pseudo-labels.

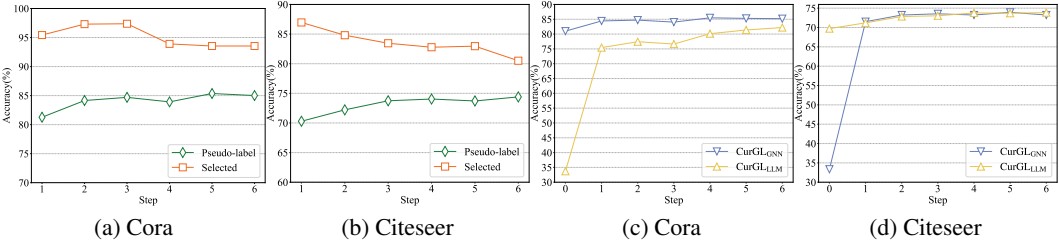

|     |     |     |     |
| --- | --- | --- | --- |
| (a) Cora | (b) Citeseer | (c) Cora | (d) Citeseer |

Figure 5: **(a)(b)**Accuracy of all pseudo-labels and selected pseudo-labels during training on Cora and Citeseer. **(c)(d)**Accuracy of **CurGL**$_{GNN}$ and **CurGL**$_{LLM}$ on the test set during training on Cora and Citeseer.

### 4.4 SELECTION STRATEGY ANALYSIS

In this section, we verify the effectiveness of our node selection strategy by comparing the accuracy of pseudo-labeled nodes selected by our method with the overall accuracy of all pseudo-labeled nodes, as shown in Figure 5 (a) and (b). (1) We observe that the accuracy of all pseudo-labeled nodes increases during training, indicating that our method improves the quality of pseudo-labels over time. (2) The accuracy of pseudo-labeled nodes selected by our strategy consistently exceeds the overall accuracy of all pseudo-labeled nodes, demonstrating that our selection strategy effectively identifies high-confident pseudo-labeled nodes for training, leading to improved performance on TAGs.

## 4.5 TRAINING DYNAMICS ANALYSIS

In this section, we analyze the training dynamics of our method by examining the accuracy of the GNN and LLM on the test set during training. Figure 5 (c) and (d) show the accuracy of GNN and LLM in **CurGL** on the test set during training. We observe that the accuracy of GNN and LLM increases over time, indicating that our method progressively aligns the LLM and GNN, balancing the learning of textual attributes and topological structures in TAGs.

## 5 RELATED WORKS

**GNN-LLM Alignment for TAGs**    TAGs have gained increasing attention in the research community, with GraphLLM (Jin et al., 2023d; Tang et al., 2024b; Xia et al., 2024; Kong et al., 2024; Li et al., 2023b; Ren et al., 2024; Zou et al., 2023; Tang et al., 2024b; Huang et al., 2024a; Chen et al., 2024b;d; Wei et al., 2024a; Guo et al., 2024) methods emerging as a promising approach to leverage the strengths of both LLMs and GNNs. Among these GraphLLM methods, the alignment between LLMs and GNNs (Li et al., 2023c; Jin et al., 2023a) has attracted an increasing amount of attention by aligning them in the same vector space. For instance, GLEM (Zhao et al., 2022a) integrates GNNs and LLMs, within a variational Expectation-Maximization framework. It alternates updates between LLMs and GNNs, enhancing performance on downstream tasks. PATTON (Jin et al., 2023b) enhances GraphFormer Yang et al. (2021) by introducing two novel pre-training strategies tailored for text-attributed graphs: network-contextualized masked language modeling and masked node prediction. However, exising GNN-LLM alignment methods ignore the varying learning difficulties of textual attributes and structures across nodes in TAGs, leading to a suboptimal textual and structural representation learning. In this paper, we focus on the balance of learning difficulties between textual attributes and structures on a node-by-node basis to improve the alignment between GNNs and LLMs.

**Curriculum Learning on Graphs**    Graph Curriculum Learning (GCL) is different from traditional Curriculum Learning due to the inherent dependencies of graph data. As a result, curriculum learning methods (Gong et al., 2019; Zhou et al., 2022; Wang et al., 2023; Li et al., 2024a; 2023a) designed for independent data cannot be directly applied to graphs. Researchers leverage graph structures to measure difficulty through predefined or automated strategies. CLNode (Wei et al., 2023) is a Curriculum Graph Learning method that measures local difficulty by considering the class diversity among a node's neighbors and uses global features to identify mislabeled nodes. RCL (Zhang et al., 2023) is a Curriculum Graph Learning method that gradually integrates node relationships into the training process, based on the complexity of those relationships. TSS (Wu et al., 2024) is a distinctive perspective on employing curriculum learning methods specifically tailored for noisily labeled graphs. However, existing GCL methods lack a measurer that accounts for global structural information to estimate the learning difficulty of nodes and overlook the varying difficulty across different classes, which can lead to subgraph imbalance. Additionally, these methods ignore textual attributes learning difficulty, limiting their applicability to TAGs. In this paper, we propose a Curriculum Learning approach for node selection that considers textual attributes, topological structures, and the varying difficulty across different classes in TAGs.

## 6 CONCLUSION

In this paper, we propose a curriculum GNN-LLM alignment (**CurGL**) for TAGs, aimed at strategically balance the learning of textual attributes and topological structures of nodes, improving node representation learning. We introduce a text-structure difficulty measurer to estimate learning difficulty of textual and structural information of nodes. Then we propose a class-based node selection strategy to balance the training process by gradually scheduling more nodes, considering the varying class-specific node's difficulties. Finally, a curriculum co-play alignment by iteratively promoting useful information from LLM and GNN, progressively enhancing both components with a balanced textual and structural information. Extensive experiments on real-world show that our method outperforms existing GraphLLM approaches. In the future, we would like to extend our method to other graph-related tasks, such as link prediction and graph classification.

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

## A METHODS DISCUSSION AND IMPLIMENTATION

### A.1 DISCUSSION ON GLOBAL CENTER-BOUNDARY DETECTION

Our method is inspired by the class-conditional betweenness centrality method proposed by (Wu et al., 2024). The betweenness centrality of a node $i$ is defined as follows:

$$\mathbf{b}_i = \frac{1}{N(N-1)} \sum_{u \neq i \neq v} \frac{\sigma_{u,v}(i)}{\sigma_{u,v}} \tag{11}$$

where $\sigma_{u,v}$ denotes the number of shortest paths from $u$ to $v$, and $\sigma_{u,v}(i)$ denotes the number of shortest paths from $u$ to $v$ that pass through $i$.

Our method differs from prior methods by focusing on detecting not only boundary nodes but also center nodes. Our global center-boundary detection method is defined as follows:

$$D_s(i) = \frac{1}{N(N-1)} \left( \sum_{\substack{u \neq i \neq v \\ \mathbf{y}_u \neq \mathbf{y}_v}} \underbrace{\frac{\sigma_{u,v}(i)}{\sigma_{u,v}}}_{\text{Class Boundary}} - \gamma \sum_{\substack{u \neq i \neq v \\ \mathbf{y}_u = \mathbf{y}_i = \mathbf{y}_v}} \underbrace{\frac{\sigma_{u,v}(i)}{\sigma_{u,v}}}_{\text{Class Center}} \right) \tag{12}$$

We use two cases to illustrate the effectiveness of our methods to measure the topological structure difficulty of nodes, as shown in Figure 6. We denote the node 3 in case A as A3, and denote the node 3 in case B as B3. The first term for node A3 is the same as for node B3. So class-conditional betweenness centrality method fails to distinguish these two nodes. Our method consider the center nodes, the second term for A3 is larger than for B3 the structural difficulty of B3 is higher than that of A3 after the former minus the latter. This naturally reflects the more structural complexity of node B3.

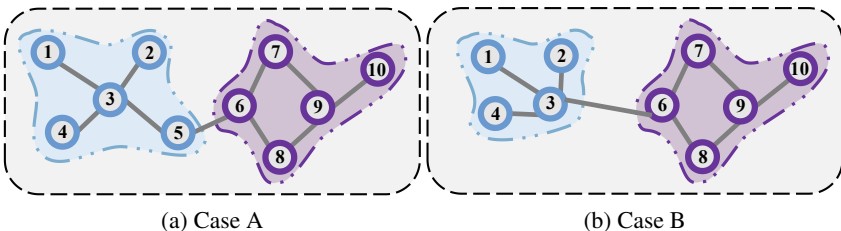

|      (a) Case A      |      (b) Case B      |

Figure 6: Two case to illustrate the difficulty considering the topological structure of nodes.

### A.2 GLOBAL CENTER-BOUNDARY DETECTION IMPLIMENTATION

Following (Wu et al., 2024), we adopt the *random walk* approach instead of calculating the shortest path to assess the topological structure difficulty of nodes, thereby avoiding the high computational cost associated with shortest path searches (Noh & Rieger, 2004; Liu & Lü, 2010; Zhao et al., 2022b). Specifically, we employ the Personalized PageRank (PPR) method (Bahmani et al., 2010; Haveliwala et al., 2003) to implement the random walk, which leads to the final formulation of our global center-boundary detection method, as defined below.

$$D_s(i) := \frac{1}{N(N-1)} \left( \sum_{\substack{u \neq i \neq v \\ \mathbf{y}_u \neq \mathbf{y}_v}} \frac{\boldsymbol{\pi}_{u,i} \boldsymbol{\pi}_{i,v}}{\boldsymbol{\pi}_{u,v}} - \gamma \sum_{\substack{u \neq i \neq v \\ \mathbf{y}_u = \mathbf{y}_i = \mathbf{y}_v}} \frac{\boldsymbol{\pi}_{u,i} \boldsymbol{\pi}_{i,v}}{\boldsymbol{\pi}_{u,v}} \right), \tag{13}$$

where $\boldsymbol{\pi}$ represents the Personalized PageRank matrix for the node, calculated as $\boldsymbol{\pi} = \alpha(\mathbf{I} - (1 - \alpha)\hat{\mathbf{A}})^{-1}$ ($\pi \in \mathbb{R}^{N \times N}$). Let $\mathbf{D} \in \mathbb{R}^{N \times N}$ represent the diagonal matrix, and $\hat{\mathbf{A}} \in \mathbb{R}^{N \times N}$ denote the normalized adjacency matrix, calculated as $\mathbf{D}^{-\frac{1}{2}} \mathbf{A} \mathbf{D}^{-\frac{1}{2}}$. The element $\pi_{u,v}$ denotes the probability of reaching node $v$ from node $u$ during an $\alpha$-random walk. $\mathbf{y}$ represents the label of the node, and $\gamma$ is a hyperparameter that controls the balance between center and boundary.

### A.3 PACING FUNCTION IMPLIMENTATION

We propose a class-based node selection strategy. This approach selects nodes based on their class-specific difficulty levels while maintaining sampled subgraph class balance. Specifically, we divide the candidate nodes $\mathcal{V}_C$ into $K$ classes, denoted as $\mathcal{V}_C = [\mathcal{V}_{C_1}, \mathcal{V}_{C_2}, \ldots, \mathcal{V}_{C_K}]$, where $\mathcal{V}_{C_k}$ represents the candidate nodes in class $k$. We then select low-difficulty nodes from each class in a proportion to $\lambda$, which is controlled by a $\mathrm{PacingFunction}$. We utilize three distinct pacing functions: linear, root, and geometric.

- linear:

$$\lambda_t = \min(1, \lambda_0 + (1 - \lambda_0) * \frac{t}{T}) \tag{14}$$

- root:

$$\lambda_t = \min(1, \sqrt{\lambda_0^2 + (1 - \lambda_0^2) * \frac{t}{T}}) \tag{15}$$

- geometric:

$$\lambda_t = \min(1, 2^{\log_2 \lambda_0 - \log_2 \lambda_0 * \frac{t}{T}}) \tag{16}$$

where $\lambda_0 \in [0, 1]$ is the initial proportion of candidate nodes selected, $T$ is the total number of training steps, and $t \in \{1, \ldots, T\}$ is the current training step. The $\mathrm{PacingFunction}$ is a monotonically increasing function of the training step $t$, controling the proportion of nodes $\lambda_t$ selected from each class at each step.

### A.4 METHODS ALGORITHM

---

**Algorithm 3** Class-based Pseudo-label Selection

---

    **Input:** Unlabeled nodes $\mathcal{V}_U$, last model prediction $\hat{\mathbf{y}}$, current training step $t$, confidence selection hyperparameter $\alpha$.
    **Output:** Candidate nodes $\mathcal{V}_C^t$.
 1: Initialize $\mathcal{V}_C^t = \mathcal{V}_L$
 2: Devide $\mathcal{V}_U$ into $K$ classes, $\mathcal{V}_U = [\mathcal{V}_{U_1}, \mathcal{V}_{U_2}, \ldots, \mathcal{V}_{U_K}]$
 3: **for** $k = 1$ to $K$ **do**
 4:     Sort $\mathcal{V}_{U_k}$ based on $\hat{\mathbf{y}}$ in descending order
 5:     Select $\alpha$ of nodes from $\mathcal{V}_{U_k}$ and add them to $\mathcal{V}_C^t$
 6: **end for**

---

## B RELATED WORKS

### B.1 GRAPHLLM FOR TAGS

**LLM-as-Enhancer.** These methods utilizes LLMs to enrich initial node embeddings in GNNs with semantic knowledge relevant to the nodes (Xie et al., 2023; Jin et al., 2023e; Qian et al., 2023; Huang et al., 2023; Zhu et al., 2024a; Jin et al., 2023c; Wei et al., 2024a; Tan et al., 2024; He & Hooi, 2024; Pan et al., 2024). For example, SimTeG (Duan et al., 2023) applies LoRA (Hu et al., 2021) to fine-tune LLMs on graph text corpora in a parameter-efficient manner, and subsequently leverages the fine-tuned LLM to generate node representations for GNN predictions.

**LLM-as-Predictor.** These methods employ LLMs as predictors within a unified generative framework for graph tasks (Wang et al., 2024a; Guo et al., 2023; Zhao et al., 2023a; Liu et al., 2024; Liu & Wu, 2023; Fatemi et al., 2023; Hu et al., 2023; Liu et al., 2023b; Shi et al., 2023; Qin et al., 2023; Das et al., 2023; Cao et al., 2023; Ai et al., 2023; Perozzi et al., 2024; Wei et al., 2024b; Wang et al., 2024b). For example, InstructGLM (Ye et al., 2023) employs a generative framework in which LLMs are trained to predict node labels by generating them based on the nodes' textual attributes. On the other hand, GraphGPT (Tang et al., 2024a) adapts LLMs for downstream graph tasks using instruction tuning, leveraging natural language alongside a graph-text aligner to capture and convey the structural information of the graph.

**GNN-LLM Alignment.** GNN-LLM alignment ensure that the distinct strengths of each encoder are maintained, while aligning their embedding spaces at a designated stage (Chandra et al., 2020; Edwards et al., 2021; Sánchez et al., 2022; Su et al., 2022; Zhao et al., 2022a; Mavromatis et al., 2023; Brannon et al., 2023; Wen & Fang, 2024). For example, GLEM (Zhao et al., 2022a) integrates GNNs and LLMs within an EM framework, where the two models iteratively generate pseudo-labels to assist each other. In contrast, PATTON (Jin et al., 2023b) introduces a novel pretraining framework for language models on text-rich networks, combining network-contextualized masked language modeling with masked node prediction to enhance performance on tasks that involve both textual and structural data.

## C EXPERIMENTS DETAILS

### C.1 BASELINES

We adopt several representative GNNs, Curriculum Graph Learning, Pretrained Language Models, and GraphLLM methods as baselines to compare with our approach on real-world datasets. We categorize the GraphLLM methods into three groups based on their training strategies (Jin et al., 2023a; Li et al., 2023c). A more detailed description of these baselines is provided as follows:

- **GNNs**
    - **GCN.** (Kipf & Welling, 2017) Graph Convolutional Networks (GCN) is a well-known GNN model that updates node representations by aggregating information from their neighboring nodes.
    - **GAT.** (Velickovic et al., 2017) Graph Attention Networks (GAT) is a graph neural network that uses attention mechanisms to weigh the importance of neighboring nodes when updating node representations.
    - **GraphSAGE.** (Hamilton et al., 2017) GraphSAGE is a model for inductive representation learning on large graphs, which generates node embeddings by sampling and aggregating features from neighboring nodes.

- **Curriculum Graph Learning**
    - **CLNode.** (Wei et al., 2023) Curriculum Learning for Node Classification (CLNode) is a Curriculum Graph Learning method that measures local difficulty by considering the class diversity among a node's neighbors and uses global features to identify mislabeled nodes.
    - **RCL.** (Zhang et al., 2023) Relational Curriculum Learning (RCL) is a Curriculum Graph Learning method that gradually integrates node relationships into the training process, based on the complexity of those relationships.
    - **TSS.** (Wu et al., 2024) Topological Sample Selection (TSS) is a distinctive perspective on employing curriculum learning methods specifically tailored for noisily labeled graphs.

- **Pretrained Language Models**
    - **BERT.** (Kenton & Toutanova, 2019) Bidirectional Encoder Representations from Transformers (BERT) is a pre-trained transformer model that uses a bidirectional attention mechanism to understand the context of words in a sentence, enabling it to excel in various natural language processing tasks.
    - **Sent-BERT.** (Reimers & Gurevych, 2019) Sentence-BERT is an adaptation of the BERT model specifically designed to generate sentence embeddings, allowing for effective semantic similarity comparisons and improved performance on tasks like clustering and information retrieval.
    - **RoBERTa.** (Liu et al., 2019) RoBERTa is an enhanced version of BERT that modifies the training process by using more data, longer training times, and removing the next sentence prediction objective to improve overall model performance.

- **LLM as Enhancer**
    - **GIANT.** (Chien et al., 2022) GIANT introduces a method that utilizes XR-Transformers Zhang et al. (2021) to perform neighborhood prediction, resulting in a large language model (LLM) capable of generating feature vectors that surpass those produced by both

bag-of-words and standard BERT Kenton & Toutanova (2019) embeddings for node classification tasks.

- **TAPE.** (He et al., 2023) TAPE employs custom prompts to engage LLMs, producing predictions and textual explanations for each node. These text explanations are then refined using DeBERTa He et al. (2020) to transform them into node embeddings suitable for GNNs. Consequently, GNNs leverage a blend of original text features, explanation-derived features, and prediction features to accurately predict node labels.

- **OFA.** (Liu et al., 2023a) OFA represents all nodes and edges using human-readable texts and encodes these elements from various domains into a unified space using LLMs. The framework is designed to be adaptable for different tasks by incorporating task-specific prompting substructures into the input graph.

- **ENGINE.** (Zhu et al., 2024b) ENGINE incorporates a tunable G-Ladder module into each layer of the LLM, which employs a message-passing mechanism to integrate structural information. This configuration allows the output from each LLM layer, specifically token-level representations, to be transferred to the corresponding G-Ladder module. Here, the node representations are refined and subsequently utilized for node classification tasks.

• **LLM as Predictor**

- **InstructGLM.** (Ye et al., 2023) InstructGLM proposes templates that encapsulate the local ego-graph structure, covering up to a 3-hop connection for each node, and utilizes instruction tuning to enhance node classification tasks.

- **GraphText.** (Zhao et al., 2023b) GraphText incorporates a fusion module that merges the structural representations derived from GNNs with the contextual hidden states from LLMs, such as the encoded node text. This integration allows the structural data from the GNN adapter to enhance the textual information from the LLMs, creating a cohesive representation suitable for supervised training.

- **GraphAdapter.** (Huang et al., 2024b) GraphAdapter introduces a method where the graph encoder is first aligned with natural language semantics through text-graph grounding. Subsequently, the trained graph encoder is integrated with a LLM using a projector. By employing a two-stage instruction tuning process, the model is enabled to directly address graph-related tasks using natural language, thereby achieving zero-shot transferability.

- **LLaGA.** (Chen et al., 2024a) LLaGA employs node-level templates to transform graph data into structured sequences that are subsequently embedded into token space. This mapping enables LLMs to handle graph-structured data, improving their versatility, generalizability, and interpretability.

• **LLM GNN Alignment**

- **GLEM.** (Zhao et al., 2022a) GLEM integrates GNNs and LLMs, within a variational Expectation-Maximization (EM) framework. It alternates updates between LLMs and graph neural networks (GNNs) during the E-step and M-step, enhancing performance on downstream tasks.

- **PATTON.** (Jin et al., 2023b) PATTON enhances GraphFormer Yang et al. (2021) by introducing two novel pre-training strategies tailored for text-attributed graphs: network-contextualized masked language modeling and masked node prediction.

## C.2 DATASETS

Here we show the statistics of the datasets used in our experiments in Table 2 and briefly introduce these real-world datasets as follows:

- **Cora.** (Yang et al., 2016) is a citation network focuses on computer science research papers. Each node in the network corresponds to a paper, which includes titles and abstracts as text attributes. Edges in this network represent citation links between the papers. The dataset categorizes each paper, and the raw text data is available through the GitHub repository mentioned in Chen et al. Chen et al. (2024c).

Table 2: Statistics of all datasets, including the number of nodes, edges, average degree, average number of tokens per node, number of classes, the percentage of training/validation/testing nodes, amd the type of node text and domain.

| Dataset | # Nodes | # Edges | Avg. # Deg | Avg. # Tok | # Classes | # Train/Val/Test | Node Text | Domain |
|---------|---------|---------|-----------|-----------|-----------|------------------|-----------|--------|
| Cora | 2,708 | 5,429 | 4.01 | 186.53 | 7 | 5.17/18.46/76.37% | Paper content | Citation |
| Citeseer | 3,186 | 4,277 | 2.68 | 213.16 | 6 | 3.77/15.69/80.54% | Paper content | Citation |
| Pubmed | 19,717 | 44,338 | 4.50 | 468.56 | 3 | 0.30/2.54/97.16% | Paper content | Citation |
| Ogbn-arxiv | 169,343 | 1,166,243 | 13.77 | 243.19 | 40 | 53.70/17.60/28.70% | Paper content | Citation |
| WikiCS | 11,701 | 216,123 | 36.94 | 642.04 | 10 | 4.96/15.12/49.97% | Entity description | Web link |

- **Citeseer.** (Yang et al., 2016) is a citation network comprising research papers in the field of computer science. It features text attributes for 3,186 nodes, with each node representing a paper and each edge indicating a citation relationship between two papers. The raw text data for this dataset is sourced from the GitHub repository listed in Chen et al. (Chen et al., 2024c).

- **Pubmed.** (Yang et al., 2016) is a citation network of research papers in the biomedical domain. Like the others, each node represents a paper, and each edge denotes a citation relationship between two papers. The raw text data for PubMed is sourced from the GitHub repository mentioned in Chen et al. (Chen et al., 2024c).

- **Ogbn-arxiv.** (Hu et al., 2020) is a citation network collected from the arXiv platform, comprising papers and their citation relationships. Each node represents a paper, with edges denoting citation relationships. The raw text data for Ogbn-arxiv is sourced from the GitHub repository in OFA (Liu et al., 2023a) and is directly accessible via the link provided by the Stanford Network Analysis Project (SNAP)[1].

- **WikiCS.** (Mernyei & Cangea, 2020) is an internet link network where each node represents a Wikipedia page, and each edge represents a reference link between pages. The raw text data, including the name and content of each Wikipedia entry, collected from OFA (Liu et al., 2023a). Each node's label indicates the category of the Wikipedia entry.

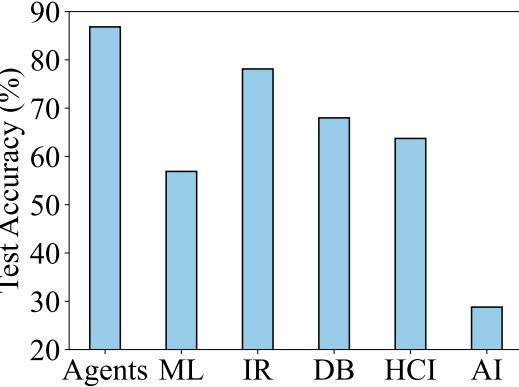

Figure 7: Test accuracy of the GNN-LLM alignment method GLEM (Zhao et al., 2022a) on the Citeseer dataset varies significantly across different classes, particularly in the ML (Machine Learning) and AI (Artificial Intelligence) categories, indicating that the learning difficulty of different classes is not uniform across classes.

## D  DIFFICULTY VARY ACROSS CLASSES

We also observe that the text and structure learning difficulty can vary significantly across different classes of nodes. For instance, in citation networks, papers in fields such as machine learning and artificial intelligence may share similar keywords in their textual attributes and exhibit more interactions in their topological structures, making it more challenging to distinguish between these two classes, as shown in Figure 7.

---

[1] https://snap.stanford.edu/ogb/data/misc/ogbn_arxiv

# E  LOW-QUALITY EXPERIMENT

**Experimental Setting.** To evaluate the robustness of our method in low-quality label scenarios, we conduct experiments on synthetic noisy datasets. We introduce random noise at varying rates to the labels of nodes in the training and validation datasets to simulate noisy conditions. We compare our method against GNNs, curriculum graph learning methods, and competitive GraphLLM methods across different levels of synthetic noise. The results are presented in Figure 8.

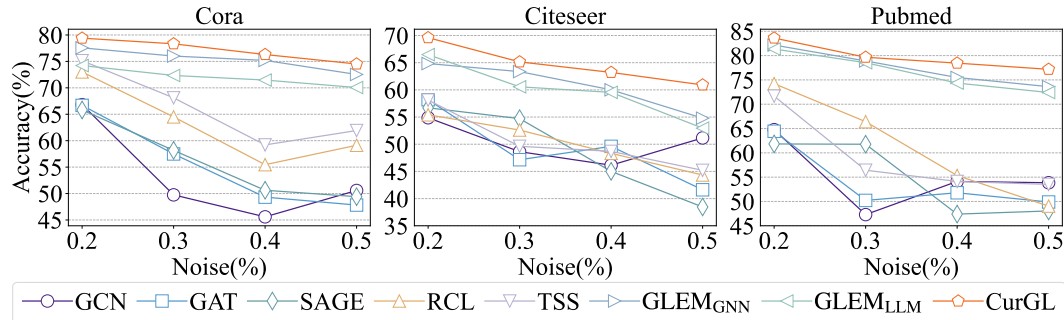

Figure 8: Performance (%) of different methods on synthetic noisy datasets with different noise levels.

**Exprimental Results.** Based on the results in Figure 8, we make the following observations: (1) Competitive GraphLLM methods, while effective on real-world datasets, experience a significant drop in performance on synthetic noisy datasets as noise rate increase. This suggests that GraphLLM methods are sensitive to noise and struggle to learn accurate node representations in noisy environments. (2) Our method achieves the better performance than GNNs, curriculum graph earning and GraphLLM methods on all synthetic noisy datasets. This demonstrates that our method is robust to noisy scenarios in TAGs.

# F  TIME EFFICIENCY

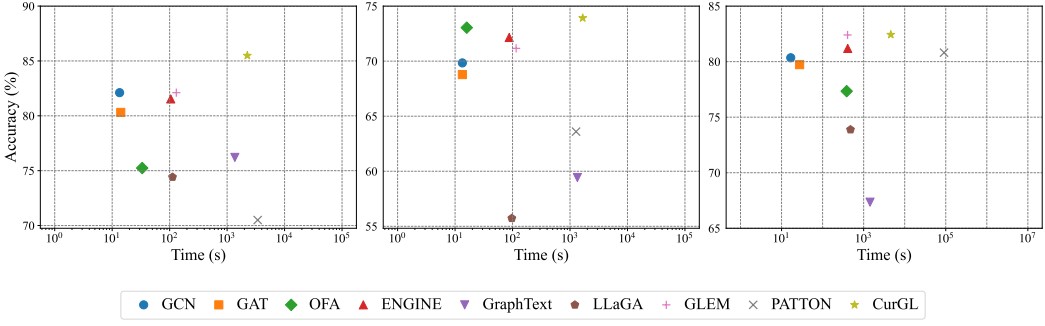

Figure 9: Training time analysis on Cora, Citeseer and WikiCS.

The experimental results show that our method achieves a significant improvement over the baselines, with an acceptable time consumption, and some baselines even require more time than our method.

# G  PROBLEM SHOWCASE

The results after training separately with the LLM and GNN are shown in the figure 10. Experimental findings reveal that some nodes are easily predicted by the LLM but are challenging for the GNN, indicating that the textual content of these nodes is relatively simple. Conversely, some nodes are

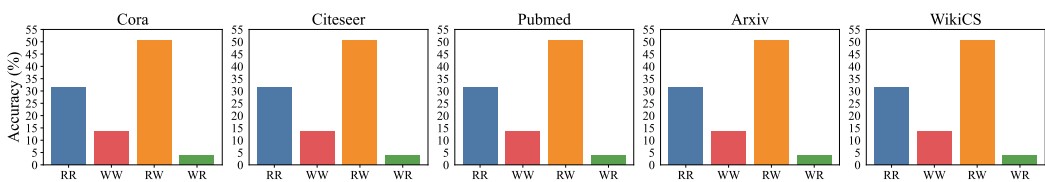

Figure 10: *RR* represents the ratio of nodes where both the LLM and GNN make correct predictions. WW represents the ratio of nodes where both the LLM and GNN make incorrect predictions. RW represents the ratio of nodes where the LLM makes a correct prediction while the GNN makes an incorrect prediction. WR represents the ratio of nodes where the GNN makes a correct prediction while the LLM makes an incorrect prediction.

easily predicted by the GNN but misclassified by the LLM, suggesting that the structural features of these nodes are easier to learn.

## H  DERIVATION OF LOSS FUNCTIONS IN THE VARIATIONAL EM ALGORITHM

In this section, we provide a derivation of the loss functions for our proposed algorithm, which combines a LLM and a GNN within a variational EM framework following (Zhao et al., 2022a). We ensure that the final loss functions match those defined in Equations equation 9 and equation 10.

### H.1  PROBLEM SETUP

We consider a graph $\mathcal{G} = (\mathcal{V}, \mathcal{E})$, where the node set $\mathcal{V}$ is partitioned into:

- A labeled set $\mathcal{V}_L$ with known labels $\mathbf{y}_L$.
- An unlabeled set $\mathcal{V}_U$ with unknown labels $\mathbf{y}_U$.

Each node $v \in \mathcal{V}$ has:

- Text features $\mathbf{s}_v$.
- An ego-graph $\mathcal{G}_v$ (for GNN input).

Our goal is to predict the labels of the unlabeled nodes $\mathbf{y}_U$ by leveraging both the text features and the graph structure.

### H.2  PRETRAINING PHASE

We begin by pretraining:

1. **LLM Pretraining** ($f^p$): Train the initial LLM $f^p$ on the labeled node set $\mathcal{V}_L$, using $\mathbf{s}_v$ as input and $\mathbf{y}_v$ as target, to obtain initial text embeddings $\mathbf{h}_v$.
2. **GNN Pretraining** ($g^p$): Train the initial GNN $g^p$ on $\mathcal{V}_L$, using $\mathbf{h}_v$ as input and $\mathbf{y}_v$ as target, to generate initial pseudo-labels $\hat{\mathbf{y}}_v$ for $v \in \mathcal{V}_U$.

The models to be optimized are:

- LLM $f_\theta$ with parameters $\theta$.
- GNN $g_\phi$ with parameters $\phi$.

### H.3  VARIATIONAL EM FRAMEWORK

We aim to maximize the log-likelihood of the observed labels:

$$\log p(\mathbf{y}_L | \mathbf{s}_V, A) = \log \sum_{\mathbf{y}_U} p(\mathbf{y}_L, \mathbf{y}_U | \mathbf{s}_V, A). \tag{17}$$

Due to the intractability of summing over all possible $\mathbf{y}_U$, we introduce a variational distribution $q(\mathbf{y}_U | \mathbf{s}_U)$ and derive an evidence lower bound (ELBO):

$$\log p(\mathbf{y}_L | \mathbf{s}_V, A) \geq \mathbb{E}_{q(\mathbf{y}_U | \mathbf{s}_U)} \left[ \log p(\mathbf{y}_L, \mathbf{y}_U | \mathbf{s}_V, A) - \log q(\mathbf{y}_U | \mathbf{s}_U) \right]. \tag{18}$$

Our optimization involves alternating between:

- **E-step**: Optimize $q$ (the LLM model) given $p$ (the GNN model).
- **M-step**: Optimize $p$ (the GNN model) given $q$ (the LLM model).

### H.4 PARAMETERIZATION

We parameterize:

- **Variational Distribution** $q$: Modeled by the LLM $f_\theta$, predicting labels based on text features $\mathbf{s}_v$:

$$q_\theta(\mathbf{y}_U | \mathbf{s}_U) = \prod_{v \in \mathcal{V}_U} q_\theta(\mathbf{y}_v | \mathbf{s}_v). \tag{19}$$

- **Model Distribution** $p$: Modeled by the GNN $g_\phi$, predicting labels based on node embeddings $\mathbf{h}_v$, graph structure, and labels of neighboring nodes:

$$p_\phi(\mathbf{y}_v | \mathbf{h}_V, A, \mathbf{y}_{V \setminus v}) = p_\phi(\mathbf{y}_v | \mathcal{G}_v, \mathbf{y}_{V \setminus v}). \tag{20}$$

Here, $\mathbf{h}_V$ are the text embeddings for all nodes, obtained from the LLM.

### H.5 E-STEP: LLM OPTIMIZATION

In the E-step, we fix the GNN parameters $\phi$ and optimize the LLM parameters $\theta$.

#### H.5.1 OBJECTIVE FUNCTION

We aim to minimize the KL divergence between $p$ and $q$:

$$\theta^{(t+1)} = \arg\min_\theta \text{KL}\left(p_\phi(\mathbf{y}_U | \mathbf{y}_L, \mathbf{h}_V, A) \,\|\, q_\theta(\mathbf{y}_U | \mathbf{s}_U)\right). \tag{21}$$

Expanding the KL divergence:

$$\begin{aligned}
\text{KL}\left(p_\phi \,\|\, q_\theta\right) &= \sum_{\mathbf{y}_U} p_\phi(\mathbf{y}_U | \mathbf{y}_L, \mathbf{h}_V, A) \log \frac{p_\phi(\mathbf{y}_U | \mathbf{y}_L, \mathbf{h}_V, A)}{q_\theta(\mathbf{y}_U | \mathbf{s}_U)} \\
&= -\sum_{\mathbf{y}_U} p_\phi(\mathbf{y}_U | \mathbf{y}_L, \mathbf{h}_V, A) \log q_\theta(\mathbf{y}_U | \mathbf{s}_U) + \text{const.}
\end{aligned} \tag{22}$$

Assuming independence in $q_\theta$ and using the approximation with pseudo-labels $\hat{\mathbf{y}}_v$ from the GNN, we have:

$$\text{KL}\left(p_\phi \,\|\, q_\theta\right) \approx -\sum_{v \in \mathcal{V}_U} \log q_\theta(\hat{\mathbf{y}}_v | \mathbf{s}_v) + \text{const.} \tag{23}$$

### H.5.2 FINAL LOSS FUNCTION

Including the supervised loss on labeled data, the overall LLM loss function becomes:

$$\min_{\theta} \mathcal{L}_{\text{LLM}}(\theta) = \sum_{v \in \mathcal{V}_L} \mathcal{L}\left(f_\theta(\mathbf{s}_v), \mathbf{y}_v\right) + \sum_{v \in \mathcal{V}_U} \mathcal{L}\left(f_\theta(\mathbf{s}_v), \hat{\mathbf{y}}_v\right), \tag{24}$$

where $\mathcal{L}$ is the cross-entropy loss function, $\mathbf{y}_v$ is the true label for labeled nodes, and $\hat{\mathbf{y}}_v$ is the pseudo-label from the GNN for unlabeled nodes.

This matches the loss function in Equation equation 9.

After optimizing $\theta$, we update the text embeddings $\mathbf{h}_v = f_\theta(\mathbf{s}_v)$ and generate new pseudo-labels for the unlabeled nodes based on the LLM's output logits.

### H.6 M-STEP: GNN OPTIMIZATION

In the M-step, we fix the LLM parameters $\theta$ and optimize the GNN parameters $\phi$.

### H.6.1 OBJECTIVE FUNCTION

We aim to maximize the expected complete-data log-likelihood:

$$\phi^{(t+1)} = \arg\max_{\phi} \mathbb{E}_{q_\theta(\mathbf{y}_U|\mathbf{s}_U)}\left[\log p_\phi(\mathbf{y}_L, \mathbf{y}_U|\mathbf{h}_V, A)\right]. \tag{25}$$

Approximating $q_\theta(\mathbf{y}_U|\mathbf{s}_U)$ using the pseudo-labels $\hat{\mathbf{y}}_v$ from the LLM, we have:

$$\mathbb{E}_{q_\theta}\left[\log p_\phi(\mathbf{y}_L, \mathbf{y}_U|\mathbf{h}_V, A)\right] \approx \sum_{v \in \mathcal{V}_L} \log p_\phi(\mathbf{y}_v|\mathcal{G}_v) + \sum_{v \in \mathcal{V}_U} \log p_\phi(\hat{\mathbf{y}}_v|\mathcal{G}_v). \tag{26}$$

### H.6.2 FINAL LOSS FUNCTION

Converting to a loss function to minimize, we get:

$$\min_{\phi} \mathcal{L}_{\text{GNN}}(\phi) = \sum_{v \in \mathcal{V}_L} \mathcal{L}\left(g_\phi(\mathcal{G}_v), \mathbf{y}_v\right) + \sum_{v \in \mathcal{V}_U} \mathcal{L}\left(g_\phi(\mathcal{G}_v), \hat{\mathbf{y}}_v\right). \tag{27}$$

This matches the loss function in Equation equation 10.

After optimizing $\phi$, we generate new pseudo-labels for the unlabeled nodes based on the GNN's output logits.

