# OpenReview forum: "Curriculum GNN-LLM Alignment for Text-Attributed Graphs"
_ICLR.cc/2025/Conference — Submitted to ICLR 2025_

### Official Review · Reviewer_s8oL · 2024-10-18

**Soundness:** 1
**Presentation:** 3
**Contribution:** 2
**Rating:** 5
**Confidence:** 3

**Summary:**

This paper presents CurGL, a curriculum-based alignment method for Text-Attributed Graphs that introduces a text-structure difficulty measurer, a class-based node selection strategy, and curriculum co-play alignment to iteratively enhance Graph Neural Networks and Large Language Models. The experiments demonstrate the superiority of the method.

**Strengths:**

- The writing is fluent and easily comprehensible.
- It is one of the earliest works to apply Curriculum Learning in the field of TAGs.
- The authors propose a class-based hard node selection strategy and a text-structure difficulty measurer, which serve as an inspiration for the community.

**Weaknesses:**

- There is a lack of explanation and mathematical proof as to why the EM algorithm is used instead of updating the model parameters of LLM and GNN simultaneously.
- The experiments in this work are insufficient, as they only include node classification experiments without covering graph classification or link prediction. Additionally, I have some concerns about the current experimental results, see below.

**Questions:**

1. Could the authors provide more insights into the selection of the $\lambda$ and $\alpha$ values? For different datasets (Cora and Citeseer in Figure 4), they seem to require different values.
2. In section 4.4, why does the accuracy of pseudo-labeled nodes selected by CurGL decrease as the number of steps increases?
3. In Section 4.5, why do both of the two components, LLM and GNN, exhibit opposite trends across different datasets?
4. In the main results, it seems that a simple GCN outperforms models like TAPE and ENGINE.

---

> ### Author Response · Authors · 2024-11-24
>
> Dear Reviewer s8oL
>
> We sincerely appreciate your valuable comments! We believe the main questions may arise from a slight misunderstanding of our experiments. We address your concerns as follows.
>
> ***
>
> > W1: "There is a lack of explanation and mathematical proof as to why the EM algorithm is used instead of updating the model parameters of LLM and GNN simultaneously."
>
> Thank you for your valuable suggestion. We have added explanations and mathematical proofs related to the EM algorithm in Appendix Section H.
>
> ***
>
> > W2: "The experiments in this work are insufficient, as they only include node classification experiments without covering graph classification or link prediction. Additionally, I have some concerns about the current experimental results, see below."
>
> Many recent studies (e.g., [1–5]) focus on learning for text-attributed graph node classification. Our method emphasizes jointly learning both textual and structural features of nodes to learn effective node representations, which could be used for a variety of applications such as node classification and link prediction. The text embeddings generated by our method (LLM component) can serve as initial features for GNNs, and our fine-tuned LLM can be transferred to other methods that use LLMs as text encoders (e.g., OFA [6]), making them adaptable to various downstream tasks such as link prediction and graph classification.
>
> ***
>
> > Q1: "Could the authors provide more insights into the selection of the and values? For different datasets (Cora and Citeseer in Figure 4), they seem to require different values."
>
>
> The hyperparameter $\lambda_0$ in Eq. 8 represents the initial selection proportion of candidate nodes, and its value is adjusted within the range $[0, 1]$. As shown in the experiments, $\lambda_0$ should be selected appropriately for each dataset. Setting $\lambda_0$ too high $\left(\lambda_0 \rightarrow 1\right)$ may result in the inclusion of too many samples initially, causing the training process to resemble standard training and neglecting the progressive nature of curriculum learning. On the other hand, setting $\lambda_0$ too low $\left(\lambda_0 \rightarrow 0\right)$ may result in the use of too few samples at the beginning, preventing the model from sufficiently learning basic features, which can adversely affect subsequent training stages.
>
> The hyperparameter $\alpha$ in Algorithm 3 controls the number of selected pseudo-labeled nodes, and we adjust its value between 0 and 1. It is also essential to choose a moderate value to introduce an appropriate number of pseudo-labels. A value that is too high $\left(\alpha \rightarrow 1\right)$ would incorporate all pseudo-labels, causing the model to overly rely on them, while a value that is too low $\left(\alpha \rightarrow 0\right)$ would make the model overly depend on the original training labels, hindering the mutual learning between the GNN and the LLM.

---

> ### Author Response · Authors · 2024-11-24
>
> ***
>
> > Q2: "In section 4.4, why does the accuracy of pseudo-labeled nodes selected by CurGL decrease as the number of steps increases?"
>
> We sincerely apologize for not providing a clear explanation of this issue in the manuscript. We greatly appreciate your valuable suggestion. Below is our explanation, which has also been added to the manuscript along with relevant analysis.
>
> Our method for calculating the accuracy of pseudo-labeled nodes is to divide the number of correctly labeled nodes among the selected $V_s$ nodes by the total number of selected $V_s$ nodes. However, as the number of selected pseudo-labeled nodes $V_s$ increases, the proportion (accuracy) tends to decrease, which is a reasonable phenomenon. For instance, if we select all pseudo-labeled nodes as $V_s$, the accuracy will approximate the overall accuracy of all pseudo-labels.
>
> **Definitions:**
>
> - $N$: Total number of pseudo-labeled nodes.
> - $C$: Total number of correctly pseudo-labeled nodes. Overall accuracy: $A_{\text{overall}} = \frac{C}{N}$.
> - $V_s$: Selected subset of pseudo-labeled nodes, with $k = |V_s|$.
> - $C_s$: Number of correctly labeled nodes in $V_s$. Accuracy for $V_s$: $A_{V_s} = \frac{C_s}{k}$.
>
> **Step 1: Adding More Nodes**
>
> - Add a subset $\Delta V$, where $|\Delta V| = m$ and $\Delta C$ are correct labels.
> - New accuracy after adding $\Delta V$:
>   $$
>   A_{V_s \cup \Delta V} = \frac{C_s + \Delta C}{k + m}.
>   $$
> - Since $\Delta C \leq m$, we have:
>   $$
>   A_{V_s \cup \Delta V} \leq \frac{C_s + m}{k + m}.
>   $$
>
> **Step 2: Convergence**
>
> - As $k \to N$, $C_s \to C$:
>   $$
>   A_{V_s} \to \frac{C}{N} = A_{\text{overall}}.
>   $$
> - This shows that $A_{V_s}$ decreases as $k$ increases, converging to $A_{\text{overall}}$.
>
> **Conclusion:**
>
> The accuracy $A_{V_s}$ decreases with the size of $V_s$ and eventually converges to $A_{\text{overall}}$, as adding more nodes includes lower-confidence samples.
>
> The iterative enhancement with confident pseudo-labels aims to progressively integrate valuable information between the LLM and GNN, enriching both components with a balanced combination of textual and structural insights. Unlike approaches that depend on random pseudo-label sampling, which can introduce substantial label noise, our curriculum strategy focuses on selecting confident pseudo-labels. This ensures more reliable mutual learning between the components and improves the overall effectiveness of the integration process.
>
> ***
>
> > Q3: "In Section 4.5, why do both of the two components, LLM and GNN, exhibit opposite trends across different datasets?"
>
> Thank you for your comment. The LLM and GNN in Figure 5 (c) and (d) exhibit a trend of simultaneous improvement during the training process. This demonstrates that our LLM and GNN can collaboratively advance through iterative training.

---

> ### Author Response · Authors · 2024-11-24
>
> ***
>
> > Q4: "In the main results, it seems that a simple GCN outperforms models like TAPE and ENGINE."
>
> To ensure a fair comparison, we follow GLbench, a widely recognized benchmark for GraphLLM, to standardize the LLMs and GNNs used in both GraphLLM baselines and our method. Specifically, in GLbench, the GNNs in the GraphLLM baselines are replaced with GraphSAGE, the LLMs in the LLM-as-predictor methods are replaced with LLaMA2-7B, the LLMs used as text encoders in the GraphLLM baselines are replaced with Sentence-BERT, and BERT is employed to process raw text into embeddings for Vanilla GNNs (e.g., GCN, GAT, and GraphSAGE), which is also adopted in our method. Since we share the same goal of ensuring a fair comparison as GLbench, we follow their setting. However, our method is not constrained by this setting. As new evaluation methods emerge in future benchmarks, our approach can be seamlessly extended to those settings to maintain fair comparisons.
>
> Our experimental results are similar to the findings of previous works, GLbench [1]. One possible reason is that GCN performs better than GraphSAGE, which causes some SOTA baselines to perform worse than GCN in this case.
>
> We truly hope we’ve answered all your questions. Thank you for your support, and we’re happy to discuss further if there’s anything we missed. Thank you again!
>
> Best regards,
>
> The authors
>
> ***
>
> Reference:
>
> [1] Li, Yuhan, et al. "Glbench: A comprehensive benchmark for graph with large language models." NeurIPS 2024.
>
> [2] Zhao, Jianan, et al. "Learning on large-scale text-attributed graphs via variational inference." ICLR 2023.
>
> [3] Chen, Zhikai, et al. "Label-free node classification on graphs with large language models (llms)." arXiv preprint arXiv:2310.04668 (2023).
>
> [4] Zhu, Yun, et al. "Efficient tuning and inference for large language models on textual graphs." CoRR 2024.
>
> [5] He, Xiaoxin, et al. "Harnessing explanations: Llm-to-lm interpreter for enhanced text-attributed graph representation learning." ICLR 2024.
>
> [6] Liu, Hao, et al. "One for all: Towards training one graph model for all classification tasks." ICLR 2024.

---

> > ### Comment · Reviewer_s8oL · 2024-11-27
> >
> > Thanks for the author's detailed response, which resolves some of my confusion. I decide to raise the score to 5.

---

> > > ### Author Response · Authors · 2024-11-28
> > >
> > > We sincerely appreciate your insightful suggestion once again, which has been instrumental in refining our work. Please feel free to let us know if you have any additional concerns!

---

> ### Author Response · Authors · 2024-11-27
> **Kind Reminder**
>
> Dear s8oL,
>
> I hope this message finds you well. I am writing to kindly remind you about providing your response to the ongoing review discussion for our ICLR submission titled **”Curriculum GNN-LLM Alignment for Text-Attributed Graphs”**.
>
> We deeply value your insights and feedback, as they play a crucial role in shaping the quality and impact of our work. If you require any further clarification or additional information from our side to assist with your response, please do not hesitate to let us know.
>
> We appreciate your time and effort in contributing to the peer-review process and look forward to your valuable input.
>
> Best regards,
>
> All authors

---

### Official Review · Reviewer_ZPUW · 2024-11-03

**Soundness:** 3
**Presentation:** 3
**Contribution:** 2
**Rating:** 5
**Confidence:** 5

**Summary:**

This paper tackles the text-structure imbalance problem in representation learning on text-attributed graphs. The authors propose a curriculum GNN-LLM alignment method to balance the learning difficulties of textual and structural information.

**Strengths:**

The paper is clearly written and easy to follow.

The proposed method demonstrates improved performance.

**Weaknesses:**

The novelty of this paper is incremental. It is evident that the ideas of difficulty measurement and curriculum learning are borrowed from existing works, such as [1,2]. Additionally, using loss as a measure of learning difficulty is a well-adopted approach.

The authors emphasize the text-structure imbalance problem, i.e., the learning difficulties of different nodes vary depending on textual attributes and topological structures. However, they provide only an intuitive example in figure 1 to illustrate this issue, without presenting real experimental results to substantiate the existence of this problem.

In Table 1, it is unusual that SOTA methods perform worse than the baseline GCN. An explanation for this unexpected phenomenon is needed.

What is the computational efficiency of the proposed model? Calculating node difficulty involves computing shortest paths between nodes, which increases computational complexity and may limit the model’s applicability to large-scale datasets.

The integration of curriculum learning into the training stage is unclear. According to Algorithm 2, the training process appears closely similar to that of the existing method [2].

The mathematical representation of equations and loss functions lacks clarity. For example:

   a. In Eq. (1), what does "MLP" represent? Is it a large language model or a multi-layer perceptron?
   b. What is the relationship between $\hat{y}$ and $f_\theta(S)$? Are they equal?
   c. In Eq. (6), there is no $y_v$. What does $1$ represent? Is it a column vector or a scalar? Additionally, if $D_t$ is larger, does this imply that the loss is also larger?

[1] Mitigating label noise on graph via topological sample selection.

[2] Learning on large-scale text-attributed graphs via variational inference.

**Questions:**

Does selecting nodes with lower difficulty improve classification accuracy on nodes with higher difficulty?

---

> ### Author Response · Authors · 2024-11-24
>
> Dear Reviewer ZPUW
>
> We highly appreciate your insightful feedback for refining our work. We address your concerns as follows.
>
> ***
>
> > W1: "The novelty of this paper is incremental. It is evident that the ideas of difficulty measurement and curriculum learning are borrowed from existing works, such as [1,2]. Additionally, using loss as a measure of learning difficulty is a well-adopted approach."
>
> The previous GraphLLM paper employs the EM algorithm to jointly train LLM and GNN. However, our motivation is to address the issue of text-structure imbalance among nodes. While the EM algorithm is a classical approach, we adopt different curriculum strategies for GNN and LLM within the EM framework to tackle this imbalance. Specifically, the GNN focuses on structure-difficulty nodes, while the LLM targets text-difficulty nodes, ensuring that both models are fully trained and avoiding shortcuts.
>
> In contrast, TSS uses a single curriculum to reduce the impact of noisy labels, whereas our approach measures the structural difficulty of nodes. We have relevant discussions in Appendix section A to highlight the differences between our method and TSS, as well as the advantages of our approach.
>
> Using loss as a measure of learning difficulty is a widely adopted practice in previous research, as it is both simple and effective. We apply this established approach in our study. However, for our specific purpose of measuring the difficulty of nodes’ text, this method has not been explored before. Given that LLMs are specialized in text processing, their misjudgment of textual inputs can serve as a reliable indicator of the complexity of understanding the text. Our experiments confirm that using loss as a measure is effective in this context.
>
> ***
>
> > W2: "The authors emphasize the text-structure imbalance problem, i.e., the learning difficulties of different nodes vary depending on textual attributes and topological structures. However, they provide only an intuitive example in figure 1 to illustrate this issue, without presenting real experimental results to substantiate the existence of this problem."
>
> We appreciate your suggestion for presenting real experimental results to substantiate text-structure imbalance problem. We have added an experiment in Appendix section G to illustrate the issue of text-structure imbalance. Specifically, we trained the LLM and GNN separately and evaluated their performance on the all nodes. The results show that some nodes are correctly predicted by the LLM but incorrectly predicted by the GNN, while others are correctly predicted by the GNN but incorrectly predicted by the LLM. This demonstrates the existence of the text-structure imbalance problem.
>
>
> | Dataset   | Total Nodes | Both Correct (%) | Both Wrong (%) | GNN Correct, LM Wrong (%) | LM Correct, GNN Wrong (%) |
> |-----------|-------------|------------------|----------------|---------------------------|---------------------------|
> | Cora      | 2708        | 31.68            | 13.81          | 50.55                     | 3.95                      |
> | Citeseer  | 3186        | 29.60            | 23.76          | 41.81                     | 4.83                      |
> | Pubmed    | 19717       | 41.76            | 15.44          | 33.82                     | 8.98                      |
> | Arxiv     | 169343      | 73.44            | 16.75          | 5.60                      | 4.21                      |
> | WikiCS    | 11701       | 55.71            | 13.97          | 25.99                     | 4.32                      |
>
>
> ***
>
> > W3: "In Table 1, it is unusual that SOTA methods perform worse than the baseline GCN. An explanation for this unexpected phenomenon is needed."
>
> To ensure a fair comparison, we follow GLbench, a widely recognized benchmark for GraphLLM, to standardize the LLMs and GNNs used in both GraphLLM baselines and our method. Specifically, in GLbench, the GNNs in the GraphLLM baselines are replaced with GraphSAGE, the LLMs in the LLM-as-predictor methods are replaced with LLaMA2-7B, the LLMs used as text encoders in the GraphLLM baselines are replaced with Sentence-BERT, and BERT is employed to process raw text into embeddings for Vanilla GNNs (e.g., GCN, GAT, and GraphSAGE), which is also adopted in our method. Since we share the same goal of ensuring a fair comparison as GLbench, we follow their setting. However, our method is not constrained by this setting. As new evaluation methods emerge in future benchmarks, our approach can be seamlessly extended to those settings to maintain fair comparisons.
>
> Our experimental results are similar to the findings of previous works, GLbench [1]. One possible reason is that GCN performs better than GraphSAGE, which causes some SOTA baselines to perform worse than GCN in this case.

---

> > ### Author Response · Authors · 2024-11-29
> > **Kinder Reminder**
> >
> > Dear Reviewer ZPUW,
> >
> > I hope this message finds you well. I wanted to kindly remind you that the discussion period for our ICLR submission titled “Curriculum GNN-LLM Alignment for Text-Attributed Graphs” is approaching its conclusion.
> >
> > We truly value your insights and feedback, and your response would greatly contribute to refining and strengthening our work. If there are any remaining questions or clarifications we can address to assist with your feedback, please don’t hesitate to reach out.
> >
> > We understand how busy this time can be and sincerely appreciate the effort you invest in the review process. We look forward to your input and hope to hear from you soon.
> >
> > Best regards,
> > The authors

---

> ### Author Response · Authors · 2024-11-24
>
> ***
>
> > W4: "What is the computational efficiency of the proposed model? Calculating node difficulty involves computing shortest paths between nodes, which increases computational complexity and may limit the model’s applicability to large-scale datasets."
>
> We appreciate your suggestion for computational efficiency. We have added efficiency analysis experiments in Appendix section F. Besides, we have extended our method to large scale datasets ogbn-products.
>
> |     | ogbn-products    |
> |--------|--------|
> | LEADING   | 86.5  |
> | CurGL    | 87.1  |
>
> ***
>
> > W5: "The integration of curriculum learning into the training stage is unclear. According to Algorithm 2, the training process appears closely similar to that of the existing method."
>
>
> This algorithm utilizes a curriculum strategy to synergistically align the LLM and GNN. Initially, both models undergo pretraining on labeled data $\mathcal{V}_L$ to produce initial pseudo-labels $\hat{\mathbf{y}}$. Over the $T$ iterations, the models alternate in an EM process.
>
> In each iteration, a candidate set of nodes $\mathcal{V}^t_C$ is selected, comprising both ground-truth and pseudo-labeled nodes. The difficulty of each node $D_{st}$ is then assessed, and a subset of nodes $\mathcal{V}^t_S$ is chosen based on their difficulty levels. This selection is tailored to each model’s strengths: the GNN prioritizes nodes with structural difficulty, while the LLM addresses nodes with textual difficulty. Following this, pseudo-labels $\hat{\mathbf{y}}$ are updated to reflect the refined insights gained during the step.
>
> By leveraging the complementary capabilities of the LLM and GNN, this iterative process incrementally improves the accuracy of the pseudo-labels, ultimately enhancing the performance of the overall model.
>
> ***
>
> > W6: "The mathematical representation of equations and loss functions lacks clarity."
> - a. "In Eq. (1), what does "MLP" represent? Is it a large language model or a multi-layer perceptron?"
>
> MLP in Equation (1) refers to a multilayer perceptron used for classification immediately before the Softmax layer.
>
> - b. "What is the relationship between  $ \hat{y}$ and $ f_\theta(S)$ ? Are they equal?"
>
> Yes, both $ \hat{y}$ and $ f_\theta(S)$ represents the predicted logits.
>
> - c. "In Eq. (6), there is no $ \hat{y}_v$. What does 1 represent? Is it a column vector or a scalar? Additionally, if $D_t(v)$ is larger, does this imply that the loss is also larger?"
>
> $ \hat{y}_v$ represents the output logits of node $v$ from either the LLM or GNN. The text difficult $D_t(v)$ is measured as the residual between the groud truth value(1) and $ \hat{y}_v$.
>
> Thanks! We have corrected the mentioned notations and rigorously checked other notations in the revised paper.
>
> ***
>
> > Q1: "Does selecting nodes with lower difficulty improve classification accuracy on nodes with higher difficulty?"
>
> Previous studies have shown that an easy-to-hard learning strategy can significantly enhance a model’s generalization[2-4]. In our approach, we adopt a text-focused curriculum strategy for the LLM and a structure-focused curriculum strategy for the GNN. This dual curriculum enables both models to gradually develop a deeper understanding of textual and structural information, thereby reducing the risk of relying on shortcuts. Our experimental results verify the effectiveness of this approach.
>
>
> We truly hope we’ve answered all your questions. Thank you for your support, and we’re happy to discuss further if there’s anything we missed. Thank you again!
>
> Best regards,
>
> The authors
>
> ***
>
> Reference:
>
> [1] Li, Yuhan, et al. "Glbench: A comprehensive benchmark for graph with large language models." NeurIPS 2024.
>
> [2] Bengio, Yoshua, et al. "Curriculum learning." Proceedings of the 26th annual international conference on machine learning. 2009.
>
> [3] Wang, Xin, Yudong Chen, and Wenwu Zhu. "A survey on curriculum learning." IEEE transactions on pattern analysis and machine intelligence 44.9 (2021): 4555-4576.
>
> [4] Li, Haoyang, Xin Wang, and Wenwu Zhu. "Curriculum graph machine learning: A survey." arXiv preprint arXiv:2302.02926 (2023).

---

> > ### Comment · Reviewer_ZPUW · 2024-12-02
> >
> > Thank you for taking the time to address my questions. I have revisited both the authors’ response and the paper. However, I find the responses rather cursory, with several of my concerns not adequately addressed. Below, I outline my remaining questions and issues:
> >
> > 1. **Regarding W2**:
> >    Why does "GNN Correct, LM Wrong" achieve 50.55%, while "LM Correct, GNN Wrong" only achieves 3.95%? This discrepancy seems unusual and requires further explanation.
> >
> > 2. **Regarding W3**:
> >    The explanation for why SOTA methods underperform compared to the baseline GCN is unconvincing. Could you clarify the datasets where GCN outperforms GraphSAGE? By how much does GCN perform better? Furthermore, if GCN performs well, what is the rationale for introducing a language model in this context?
> >
> > 3. **Regarding W4**:
> >    The proposed method involves computing the shortest paths between all nodes. Could you provide a theoretical analysis of the computational complexity? While Appendix Section F presents experimental results on efficiency, it is evident that the proposed method incurs significant computational costs. Moreover, in the ogbn-products experiments, what does "LEADING" refer to?
> >  What is the runtime of the proposed method on the ogbn-products dataset?
> >
> > 4. **Regarding W5**:
> >    The iterative training stages of the proposed method bear a strong resemblance to the approach in [2]. Could you elaborate on the differences?
> >
> > 5. **Regarding W6**:
> >    If both $\hat{y}$ and $f_{\theta}(S)$ represent predicted logits, what exactly does $\theta$ denote? Does it refer to the parameters of the MLP, SeqEnc, or both? Additionally, if $D_t(v)$ is larger, does this imply that the loss is also larger?
> >
> > 6. **Regarding Q1**:
> >    The authors did not directly answer my question. Specifically, does selecting nodes with lower difficulty improve the classification accuracy of nodes with higher difficulty?
> >
> > I will consider raising my score if the authors can answer my questions well.

---

> > > ### Author Response · Authors · 2024-12-02
> > > **Authors Rebuttal(1/2)**
> > >
> > > Dear Reviewer ZPUW
> > >
> > > We highly appreciate your insightful feedback for refining our work. We further address your concerns as follows.
> > >
> > > ***
> > >
> > > > W2
> > >
> > > Thank you very much for your insightful reminder. In our experimental setup, we first use the LLM to train and predict the text of the node, and then apply the GNN, leveraging the LLM’s embeddings as the initial input. Therefore, it is reasonable for the GNN to achieve higher accuracy than the LLM, as the GNN not only utilizes the processed text but also incorporates the graph structure for prediction. The primary objective of this experiment is to demonstrate that certain nodes exhibit varying levels of difficulty for LLM and GNN to classify.
> > >
> > > ***
> > >
> > > > W3
> > >
> > > Reviewing our table, GCN consistently outperforms GraphSAGE by 1–3% across all datasets. To ensure a fair comparison of GraphLLM methods, we replaced all GNNs in the GraphLLM baselines with GraphSAGE. This might explain why GCN performs particularly well on the **Cora** dataset, surpassing almost all GraphLLM baselines. However, it is evident that GCN still falls short compared to the best GraphLLM baselines on other datasets. So TAGs need LLM/LM to be introduced.
> > > Additionally, I believe current Graph LLM research can be broadly categorized into two main directions. The first focuses on using LLMs for large-scale graph-text SFT, achieving strong performance across various domains and tasks. The second emphasizes processing graph-text data by leveraging LLMs/LMs to understand the textual modality, rather than focusing on their cross-domain and cross-task applications after large-scale pretraining. Both directions undoubtedly have their merits.
> > >
> > > Another possible reason for GCN’s relatively strong performance is that the initial textual input of the **Cora** dataset is already well-understood. As a result, adding LLM/LM enhancements may not yield significant improvements. Our experimental results are consistent with observations in previous studies[1].
> > >
> > >
> > >
> > > ***
> > >
> > > > W4
> > >
> > > We give a time complexity of the structure  difficult  measurer as below:
> > >
> > > The normalized adjacency matrix $\hat{A}$ is defined as:
> > > $$
> > > \hat{A} = D^{-\frac{1}{2}} A D^{-\frac{1}{2}} \nonumber
> > > $$
> > >
> > >
> > > Computing $D^{-\frac{1}{2}}$ requires $O(N)$ since $D$ is diagonal. The matrix multiplication $D^{-\frac{1}{2}} A D^{-\frac{1}{2}}$ requires $O(N^2)$.
> > >
> > > The Personalized PageRank matrix $\pi$ is given by:
> > > $$
> > > \pi = \alpha (I - (1-\alpha) \hat{A})^{-1} \nonumber
> > > $$
> > >
> > >
> > > Computing the inverse of an $N \times N$ matrix directly has a complexity of $O(N^3)$.
> > >
> > > The term $D_s(i)$ is defined as:
> > > $$
> > > D_s(i) = \frac{1}{N(N-1)} \left( \sum_{\substack{u \neq i \\ y_u \neq y_v}} \frac{\pi_{u,i} \pi_{i,v}}{\pi_{u,v}} - \gamma \sum_{\substack{u \neq i \\ y_u = y_v \\ y_i \neq y_v}} \frac{\pi_{u,i} \pi_{i,v}}{\pi_{u,v}} \right)  \nonumber
> > > $$
> > >
> > >
> > > For each $i$, the double summation over $u$ and $v$ requires $O(N^2)$ operations. Computing $D_s(i)$ for all $i \in \{1, \dots, N\}$ results in $O(N^3)$ total complexity.
> > >
> > > The overall time complexity is dominated by the computation of the Personalized PageRank matrix and the double summation in $D_s(i)$. Therefore, the total complexity is  $ O(N^3) $.
> > >
> > > Apologies for not clarifying LEADING**[3] earlier. It is a baseline method suggested by another reviewer, S9p5, for comparing the effectiveness of approaches on large-scale datasets. The runtime of the proposed method on the ogbn-products dataset is 17,734 seconds.

---

> > > ### Author Response · Authors · 2024-12-02
> > > **Authors Rebuttal(2/2)**
> > >
> > > ***
> > >
> > > > W5
> > >
> > > The previous GraphLLM(e.g. GLEM[2]) paper employs the EM algorithm for jointly training LLM and GNN. However, our motivation lies in addressing the issue of text-structure imbalance in node representation learning. While the EM algorithm serves as a classical approach, our Curriculum Co-play Alignment method introduces the following unique aspects:
> > >
> > > 1. **Curriculum strategies tailored to text-structure imbalance**: To tackle the imbalance issue, we design different curriculum strategies for the GNN and LLM within the EM framework. Specifically, the GNN processes structure-difficulty nodes progressively, from easy to hard, while the LLM focuses on text-difficulty nodes. This ensures both components undergo comprehensive training and avoid shortcuts.
> > > 2. **Iterative enhancement with confident pseudo-labels**: A key objective is to iteratively integrate valuable information between the LLM and GNN, progressively enriching both components with a balance of textual and structural insights. Unlike methods that rely on random pseudo-label sampling often introducing significant label noise, our curriculum strategy emphasizes the selection of confident pseudo-labels to facilitate mutual learning between the components. As shown in Figures 5(a) and 5(b), our method effectively selects high-confidence pseudo-labels.
> > >
> > >
> > >
> > > ***
> > >
> > > > W6
> > >
> > > $\theta$ denotes the parameters of the LLM (including both MLP and SeqEnc). Yes, if $D_t(v)$ is larger, it implies that the loss is also larger.
> > >
> > >
> > >
> > > ***
> > >
> > > > Q1
> > >
> > > Apologies for the earlier response. What we intended to convey is that selecting nodes with lower difficulty does not improve the classification accuracy of nodes with higher difficulty. However, an easy-to-hard learning strategy can enhance the accuracy for high-difficulty nodes.
> > >
> > > ***
> > >
> > > [1] Li, Yuhan, et al. "Glbench: A comprehensive benchmark for graph with large language models." NeurIPS 2024.
> > >
> > > [2] Zhao, Jianan, et al. "Learning on large-scale text-attributed graphs via variational inference." ICLR 2023.
> > >
> > > [3] Xue, Rui, et al. "Efficient End-to-end Language Model Fine-tuning on Graphs." arXiv preprint arXiv:2312.04737 (2023).
> > >
> > > ***
> > >
> > > Best regards,
> > >
> > > The authors

---

> ### Author Response · Authors · 2024-11-27
> **Kind Reminder**
>
> Dear ZPUW,
>
> I hope this message finds you well. I am writing to kindly remind you about providing your response to the ongoing review discussion for our ICLR submission titled **”Curriculum GNN-LLM Alignment for Text-Attributed Graphs”**.
>
> We deeply value your insights and feedback, as they play a crucial role in shaping the quality and impact of our work. If you require any further clarification or additional information from our side to assist with your response, please do not hesitate to let us know.
>
> We appreciate your time and effort in contributing to the peer-review process and look forward to your valuable input.
>
> Best regards,
>
> All authors

---

### Official Review · Reviewer_S9p5 · 2024-11-03

**Soundness:** 1
**Presentation:** 3
**Contribution:** 2
**Rating:** 3
**Confidence:** 4

**Summary:**

The paper introduces a method called Curriculum GNN-LLM Alignment designed for Text-Attributed Graphs (TAGs). The goal is to address the text-structure imbalance problem in TAGs, where nodes have varying levels of difficulty in learning textual and structural information. The CurGL approach progressively balances these difficulties through three key modules.

**Strengths:**

1.	The integration of node difficulty into node classification is clearly motivated and engaging.
2.	The presentation is well-organized, and the ablation study clearly demonstrates the impact of each technique.

**Weaknesses:**

1.	My primary concern is whether this method is effective for heterophilic graphs, as the proposed difficulty measurer seems tailored to homophilous graphs.

2.	The class-based node selection assumes balanced class sampling within each subgraph. However, this assumption may be problematic for highly imbalanced datasets, potentially leading to biased or suboptimal node selection.

3.	Related to the above, it’s unclear how the authors handle edges between selected nodes in the class-based node selection. In each subgraph, nodes from different classes are sampled proportionally, which does not guarantee connectivity among them.

4.	Another major concern is computational overhead. This method involves additional steps beyond GLEM, which already has considerable running time[1]. Efficiency analysis is essential and appears missing from the experimental section.

5.	The performance appears insufficiently strong; for instance, the results on Arxiv are weaker than those of methods like LEADING[2]. Larger datasets, such as ogbn-products and ogbn-papers100M should also be included.

6.	The paper’s claim of using LLMs seems overstated, as small language models are used. To solve this obvious mismatch and align claims with experiments , it would be more appropriate to use models like LLAMA or GPT.

7.	It is unclear if this approach generalizes to tasks beyond node classification, such as link prediction and graph classification. Focusing solely on node classification may limit the overall contribution.

8. I am curious why GLEM achieves strong performance on Cora and Pubmed, which have very few labels. The results seem inconsistent with findings from existing works[1].

9.  Similar to GLEM, this method relies heavily on the quality of pseudo-labels. Although an ablation study is provided, it does not fully address potential issues that could arise with low labeling ratios.

[1] Exploring the Potential of Large Language Models (LLMs) in Learning on Graphs

[2] Efficient End-to-end Language Model Fine-tuning on Graphs

**Questions:**

1. How does the implementation achieve sampling of simpler nodes first?
2. Are the pretrained language models in the third block in Table 1 fine-tuned with labels?

---

> ### Author Response · Authors · 2024-11-24
>
> Dear Reviewer  **S9p5**
>
> We highly appreciate your insightful feedback for refining our work. We address your concerns as follows. Please feel free to correct us if otherwise!
>
> ***
>
> > W1: "My primary concern is whether this method is effective for heterophilic graphs, as the proposed difficulty measurer seems tailored to homophilous graphs."
>
> We appreciate your valuable insights. Currently, most text-attributed graph datasets are homophilous graphs, and our method is designed based on this assumption. The core idea of our structure difficulty measure is to evaluate whether a node lies near the class boundary or at the class center to measure its learning difficulty. For heterophilic graphs, such as sequential recommendation datasets, where edges are more likely to connect nodes from different classes, we can hypothesize that nodes near the class boundary are easier to learn due to their connections to different types of nodes, while nodes at the class center are more challenging to learn. Thus, our method can be naturally extended to heterophilic graphs by leveraging this assumption.
>
> ***
>
> > W2: "The class-based node selection assumes balanced class sampling within each subgraph. However, this assumption may be problematic for highly imbalanced datasets, potentially leading to biased or suboptimal node selection."
>
> The purpose of class-based node selection is to incorporate difficulty considerations when selecting nodes based on their classes. For datasets with class imbalance, we can address this issue by modifying the original sampling strategy from selecting nodes in proportion to their class sizes to selecting an equal number of nodes from each class. This adjustment effectively mitigates the problem of class imbalance among nodes.
>
> Besides, the class imbalance setting for node classification typically involves an imbalanced distribution of classes in the training set but an equal number of nodes in each class for the validation and testing set(e.g. [1]). However, our method can select nodes from the entire dataset, including the training, validation, and testing sets, which significantly mitigates the issue of class imbalance in the training set.
>
> ***
>
> > W3: "Related to the above, it’s unclear how the authors handle edges between selected nodes in the class-based node selection. In each subgraph, nodes from different classes are sampled proportionally, which does not guarantee connectivity among them."
>
> Thank you for your valuable suggestions. We have added the explanations about edges into the revised version of our paper.
> First, we measure the text-structure difficulty of all nodes. Then, we select a certain proportion of nodes from each category. Next, we retain the edges connecting these nodes from various classes to form a subgraph. However, we cannot guarantee that these nodes will be connected, as this depends on whether there are existing edges between them in the original graph.
>
>
> ***
>
> > W4: "Another major concern is computational overhead. This method involves additional steps beyond GLEM, which already has considerable running time. Efficiency analysis is essential and appears missing from the experimental section."
>
> We appreciate your suggestion for cmputational efficiency. Computational efficiency is indeed a critical aspect. We compare the time efficiency of our method with the baselines and included additional experimental analyses in the Appendix F. The experimental results show that our method achieves a significant improvement over the baselines, with an acceptable time consumption.
>
> ***
>
> > W5: "The performance appears insufficiently strong; for instance, the results on Arxiv are weaker than those of methods like LEADING. Larger datasets, such as ogbn-products and ogbn-papers100M should also be included."
>
> LEADING[2] demonstrates powerful experimental results. We add its discussion to Related works. However, in our experimental setting, to ensure a fair comparison, we standardize the baselines by using Sentence-BERT as the text encoder for LLM-based methods and LLaMA2-7B for methods where the LLM is a predictor. Nonetheless, the LLM and GNN architectures used in LEADING differ from those in our framework, making a direct comparison inequitable. As we do not find publicly available code for LEADING, we replace the LLM and GNN in our method, achieving better results than LEADING. Additionally, we have expanded the experiment for larger datasets ogbn-products as LEADING. The detailed results are as follows.
>
> |     | ogbn-arxiv    | ogbn-products    |
> |--------|--------|--------|
> | LEADING  | 77.6  | 86.5  |
> | CurGL  | 78.1  | 87.1  |

---

> ### Author Response · Authors · 2024-11-24
>
> ***
>
> > W6: "The paper’s claim of using LLMs seems overstated, as small language models are used. To solve this obvious mismatch and align claims with experiments , it would be more appropriate to use models like LLAMA or GPT."
>
> Thank you for your valuable suggestion. In our experiments, to ensure a fair comparison, we standardize both the baselines and our method by using the same LLM and GNN, following GLBench. However, our method is inherently flexible and can be easily adapted to any LLM or GNN. Based on your insightful suggestion, we have utilized LLaMA2-7B as the LLM and obtained the following experimental results.
>
> | Model              | Cora Acc | Cora F1 | Citeseer Acc | Citeseer F1 | Pubmed Acc | Pubmed F1 | Ogbn-arxiv Acc | Ogbn-arxiv F1 | WikiCS Acc | WikiCS F1 |
> |---------------------|----------|---------|--------------|-------------|------------|-----------|----------------|---------------|------------|-----------|
> | CurGL (Llama2-7B)  | 86.65    | 84.98   | 74.59        | 69.80       | 85.82      | 85.29     | 76.89          | 61.02         | 82.88      | 81.54     |
>
>
> ***
>
> > W7: "It is unclear if this approach generalizes to tasks beyond node classification, such as link prediction and graph classification. Focusing solely on node classification may limit the overall contribution."
>
> Many recent studies (e.g., [3-7]) focus on learning for text-attributed graph node classification. Our method emphasizes jointly learning both textual and structural features of nodes to learn effective node representations, which could be used for a variety of applications such as node classification and link prediction.  The text embeddings generated by our method (LLM component) can serve as initial features for GNNs, and our fine-tuned LLM can be transferred to other methods that use LLMs as text encoders (e.g., OFA [8]), making them adaptable to various downstream tasks such as link prediction and graph classification.
>
>
> ***
>
> > W8: "I am curious why GLEM achieves strong performance on Cora and Pubmed, which have very few labels. The results seem inconsistent with findings from existing works."
>
> To ensure a fair comparison, we follow GLbench, a widely recognized benchmark for GraphLLM, to standardize the LLMs and GNNs used in both GraphLLM baselines and our method. Specifically, in GLbench, the GNNs in the GraphLLM baselines are replaced with GraphSAGE, the LLMs in the LLM-as-predictor methods are replaced with LLaMA2-7B, the LLMs used as text encoders in the GraphLLM baselines are replaced with Sentence-BERT, and BERT is employed to process raw text into embeddings for Vanilla GNNs (e.g., GCN, GAT, and GraphSAGE), which is also adopted in our method. Since we share the same goal of ensuring a fair comparison as GLbench, we follow their setting. However, our method is not constrained by this setting. As new evaluation methods emerge in future benchmarks, our approach can be seamlessly extended to those settings to maintain fair comparisons.
>
> Many previous works may have overlooked these settings for a fair comparison. Our results are consistent with the GraphLLM benchmark, GLBench[3].
>
> ***
>
> > W9: "Similar to GLEM, this method relies heavily on the quality of pseudo-labels. Although an ablation study is provided, it does not fully address potential issues that could arise with low labeling ratios."
>
> Unlike GLEM[7], it lies in addressing the issue of text-structure imbalance in node representation learning. While the EM algorithm serves as a classical approach, our Curriculum Co-play Alignment method introduces the following unique aspects:
>
> 1.	**Curriculum strategies tailored to text-structure imbalance**: To tackle the imbalance issue, we design separate curriculum strategies for the GNN and LLM within the EM framework. Specifically, the GNN processes structure-difficulty nodes progressively, from easy to hard, while the LLM focuses on text-difficulty nodes. This ensures both components undergo comprehensive training and avoid shortcuts.
>
> 2.	**Iterative enhancement with confident pseudo-labels**: A key objective is to iteratively integrate valuable information between the LLM and GNN, progressively enriching both components with a balance of textual and structural insights. Unlike methods that rely on random pseudo-label sampling—often introducing significant label noise, our curriculum strategy emphasizes the selection of confident pseudo-labels to facilitate mutual learning between the components. As shown in Figures 5(a) and 5(b), our method effectively selects high-confidence pseudo-labels.
>
> Our method focuses on selecting confident pseudo-labeled samples rather than random pseudo-labeled samples, making it more robust to low-quality labels. Comparative experiments under low-quality label conditions, as detailed in Appendix Section E, show that our approach performs effectively even when the labels are of low quality.

---

> ### Author Response · Authors · 2024-11-24
>
> | Model   | Cora  |       |       |       | Citeseer |       |       |       | Pubmed |       |       |       |
> |---------|----------------------|-------|-------|-------|-------------------------|-------|-------|-------|-----------------------|-------|-------|-------|
> |         | 0.2             | 0.3   | 0.4   | 0.5   | 0.2                 | 0.3   | 0.4   | 0.5   | 0.2              | 0.3   | 0.4   | 0.5   |
> | GCN     | 66.82               | 49.75 | 45.59 | 50.58 | 54.87                  | 48.63 | 46.14 | 51.16 | 64.77                | 47.30 | 54.12 | 53.85 |
> | GAT     | 66.68               | 57.49 | 49.32 | 47.82 | 58.18                  | 47.15 | 49.61 | 41.62 | 64.44                | 50.22 | 51.79 | 49.89 |
> | SAGE    | 65.81               | 58.17 | 50.62 | 49.41 | 56.70                  | 54.71 | 45.01 | 38.46 | 61.83                | 61.77 | 47.40 | 48.05 |
> | RCL     | 73.01               | 64.50 | 55.46 | 59.13 | 55.41                  | 52.61 | 48.32 | 44.34 | 74.23                | 66.41 | 55.34 | 49.04 |
> | TSS     | 75.13               | 68.11 | 59.20 | 61.92 | 58.12                  | 49.62 | 48.69 | 45.20 | 71.66                | 56.43 | 54.12 | 53.51 |
> | GLEM    | 77.56               | 76.01 | 75.19 | 72.53 | 64.84                  | 63.36 | 60.01 | 54.79 | 82.17                | 78.90 | 75.50 | 73.61 |
> | CurGL   | 79.40               | 78.33 | 76.30 | 74.51 | 69.60                  | 65.15 | 63.21 | 60.91 | 83.61                | 79.66 | 78.45 | 77.18 |
>
> ***
>
> > Q1: "How does the implementation achieve sampling of simpler nodes first?"
>
> We rank all nodes in ascending order based on their text-structure difficulty within each class. Then, we select the top-rank nodes from each class according to a specified percentage, as detailed in Algorithm 1.
>
> ***
>
> > Q2: "Are the pretrained language models in the third block in Table 1 fine-tuned with labels?"
>
> We use the same training set split and train the Pretrained language models on it as a text classification task.
>
> ***
>
>
> We truly hope we’ve answered all your questions. Thank you for your support, and we’re happy to discuss further if there’s anything we missed. Thank you again!
>
> Best regards,
>
> The authors
>
> ***
>
> Reference:
>
> [1] Zeng, Liang, et al. "Imgcl: Revisiting graph contrastive learning on imbalanced node classification." Proceedings of the AAAI Conference on Artificial Intelligence. Vol. 37. No. 9. 2023.
>
> [2] Xue, Rui, et al. "Efficient End-to-end Language Model Fine-tuning on Graphs." arXiv preprint arXiv:2312.04737 (2023).
>
>
> [3] Li, Yuhan, et al. "Glbench: A comprehensive benchmark for graph with large language models." NeurIPS 2024.
>
> [4] Chen, Zhikai, et al. "Label-free node classification on graphs with large language models (llms)." arXiv preprint arXiv:2310.04668 (2023).
>
> [5] Zhu, Yun, et al. "Efficient tuning and inference for large language models on textual graphs." CoRR 2024.
>
> [6] He, Xiaoxin, et al. "Harnessing explanations: Llm-to-lm interpreter for enhanced text-attributed graph representation learning." ICLR 2024.
>
> [7] Zhao, Jianan, et al. "Learning on large-scale text-attributed graphs via variational inference." ICLR 2023.
>
> [8] Liu, Hao, et al. "One for all: Towards training one graph model for all classification tasks." ICLR 2024.

---

> ### Author Response · Authors · 2024-11-27
> **Kind Reminder**
>
> Dear S9p5,
>
> I hope this message finds you well. I am writing to kindly remind you about providing your response to the ongoing review discussion for our ICLR submission titled **”Curriculum GNN-LLM Alignment for Text-Attributed Graphs”**.
>
> We deeply value your insights and feedback, as they play a crucial role in shaping the quality and impact of our work. If you require any further clarification or additional information from our side to assist with your response, please do not hesitate to let us know.
>
> We appreciate your time and effort in contributing to the peer-review process and look forward to your valuable input.
>
> Best regards,
>
> All authors

---

> > ### Comment · Reviewer_S9p5 · 2024-11-28
> >
> > Thank you for your efforts in addressing my concerns. While some issues have been resolved, a few major issues remain outstanding:
> >
> > w1. I don't quite agree with the statement: “For heterophilic graphs, such as sequential recommendation datasets, where edges are more likely to connect nodes from different classes, we can hypothesize that nodes near the class boundary are easier to learn due to their connections to different types of nodes, while nodes at the class center are more challenging to learn.” In heterophilic graphs where multiple nodes with different labels are interconnected within small diameters, clearly defining boundaries and class centers becomes complex. Additionally, many real-world graphs exhibit both homophilic and heterophilic properties, even if one is dominant. This mixed nature may challenge the proposed method’s ability to generalize effectively, potentially limiting its applicability in diverse scenarios.
> >
> > w2,w3: Thank you for addressing these points.
> >
> >
> > w4. Is the time reported in the Appendix per epoch? Could you provide the total training time until model convergence?
> > Based on my experience and the results in [1], iterative training may require significant time, which could be a notable concern regarding the practicality and efficiency of the proposed method.
> >
> > w6. I suggest revising the paper to avoid reliance on LLMs, as most experiments are conducted using light version language models. Clarifying this aspect would enhance the paper’s clarity and applicability.
> >
> > w7. I understand that due to tight rebuttal timelines, providing experimental results for link prediction may be challenging. However, including empirical results on link prediction in the revised version would strengthen the paper’s contributions.
> >
> > w8. I acknowledge that the experimental settings differ from those in [1], and the superior performance reported may be attributed to the use of a deep sentence model. Nevertheless, this does not address the underlying issue that fine-tuning the language model relies on very few labels in a non end-to-end training setup, which limits the model’s ability to incorporate sufficient knowledge. This could affect the informativeness of the language models used.
> >
> > Based on current version of this work, I feel improvements in both technical soundness and presentation are required.
> >
> > [1] Exploring the Potential of Large Language Models (LLMs) in Learning on Graphs

---

> > > ### Author Response · Authors · 2024-11-28
> > >
> > > Dear Reviewer S9p5
> > >
> > > We highly appreciate your insightful feedback for refining our work. We further address your concerns as follows.
> > >
> > > > W1
> > >
> > > We deeply respect your profound expertise in heterophilic graphs, which surpasses our current understanding. The whole idea of our work is to balance node learning by measuring the difficulty of nodes. There can be various approaches to measuring the difficulty. For instance, we could consider evaluating the heterogeneity of a node’s neighbors to determine its difficulty level. We will further explore the difficulty measurer for heterophilic graphs in our future work.
> > >
> > > We greatly appreciate your valuable suggestions. Regarding our current approach, we will incorporate discussions related to the assumption of homophilous graphs. Since most text-attributed graph datasets are homophilous, we believe our approach remains applicable in most scenarios.
> > >
> > > ***
> > > > W4
> > >
> > > No, we provide the total training time until model convergence in Appendix Section F. While iterative training does indeed require significant time, our experimental results align with your observations: methods for iterative training, such as GLEM[2] and CurGL, rank among the highest in terms of time consumption. However, our approach optimizes training time by not selecting all nodes for training in each iteration. This allows us to save considerable time during training. Moreover, as shown in Figures (c) and (d) of Section 4.5, our method achieves promising results in the early stages of training, significantly reducing time consumption, though with a slight trade-off in accuracy. This could be attributed to our ability to select representative nodes for training early on.
> > >
> > > ***
> > > > W5
> > >
> > > Thank you very much for your suggestion. In our baselines, nearly half of the models utilize LLaMA2-7b. Additionally, some earlier works with lightweight language models, such as GLEM[2] and GLBench[1], also claim to use LLMs. However, we find your suggestion highly valuable and will adopt it by replacing “LLM” with “LM” in the revised version.
> > >
> > > ***
> > > > W7
> > >
> > > Thank you very much for your understanding. Due to the limited time available during the rebuttal period, we may not be able to provide results for the link prediction task in a timely manner (as the GraphLLM baseline requires significant computational time). However, we have followed a well-known GraphLLM benchmark GLBench[1] that uses node classification as the evaluation task, which has been published at NeurIPS 2024. We believe that our method’s performance surpassing this benchmark effectively demonstrates its capability in learning node representations.
> > >
> > > ***
> > > > W8
> > >
> > > Thank you very much for your perspective. We do not believe that the fine-tuning of our LLM relies on very few labels. On the contrary, our method generates additional pseudo-labels(labeled by learned textual and structural information), enabling the LLM to learn from more nodes on the graph, including more structural information. This demonstrates that our approach can achieve strong performance even with very few labels. Furthermore, our experiments on low-quality labeled datasets also show that our method does not heavily depend on these few labels. Instead, it leverages the progressively learned textual and structural information.
> > >
> > > ***
> > > Best regards,
> > >
> > > The authors
> > >
> > > ***
> > > [1] Li, Yuhan, et al. "Glbench: A comprehensive benchmark for graph with large language models." NeurIPS 2024.
> > >
> > > [2] Zhao, Jianan, et al. "Learning on large-scale text-attributed graphs via variational inference." ICLR 2023.

---

> ### Author Response · Authors · 2024-12-02
> **Kindly Reminder**
>
> Dear Reviewer S9p5,
>
> I hope this message finds you well. I wanted to kindly remind you that the discussion period for our ICLR submission titled “Curriculum GNN-LLM Alignment for Text-Attributed Graphs” is approaching its conclusion.
>
> We truly value your insights and feedback, and your response would greatly contribute to refining and strengthening our work. If there are any remaining questions or clarifications we can address to assist with your feedback, please don’t hesitate to reach out.
>
> We understand how busy this time can be and sincerely appreciate the effort you invest in the review process. We look forward to your input and hope to hear from you soon.
>
> Best regards,
> The authors

---

### Official Review · Reviewer_t915 · 2024-11-04

**Soundness:** 2
**Presentation:** 3
**Contribution:** 2
**Rating:** 5
**Confidence:** 5

**Summary:**

The paper introduces curriculum GNN-LLM alignment to tackle the text-structure imbalance problem in learning on text-attributed graphs. It proposes a node-wise difficulty measurer, a class-based node selection strategy, and a curriculum co-play alignment to balance and enhance learning between GNNs and LLMs. Extensive experiments show CurGL outperforms selective existing models on the classic node classification task across five well-known datasets.

**Strengths:**

- The motivation to balance the role of LLMs and GNNs in a fine-grained sample-wise manner for TAG learning is straightforward and convincing.
- Writing is clear and easy to follow. Pseudo code is present to assist better understanding of the mechanisms designed.

**Weaknesses:**

- Novelty. The E-M training between LLMs and GNNs has been explored by a handful of pioneer works, e.g., SimTEG and GELM. The proposed CurGL enhances such framework by adding additional strategies for balancing the LLM and the GNN.
- Unclear and potentially unfair experimental setting for baseline llm-as-predictor models. Is is unclear that how are the llm-as-predictor models testified, especially in Table 1. Is is based on a zero-shot manner as in Glbench, or are they also trained with the same training set? In terms of training, is it training-from-scratch or fine-tuning?
- Lack of discussion of a very relevant work, GraphGPT: Graph Instruction Tuning for Large Language Models.
- The proposed CurGL relies heavily on the availability of training data, and shows poor potential in transferring learning and zero-shot generalisation. Although it leverages an LLM for textual-based learning, the design of node-wise difficulty measurer and class-based selection strategy requires high-quality labels for the model to learn and converge. While it is quite reasonable for a graph learning model, the trained CurGL cannot be adapted to unseen data (unlike LLaGA, OFA and others), making its potential application quite narrow.
- Can CurGL generalise beyond node classification, to link prediction or graph classification?

**Questions:**

Please see weaknesses.

---

> ### Author Response · Authors · 2024-11-24
>
> Dear Reviewer  **t915**
>
> We appreciate your efforts and insightful feedback, which have been instrumental in refining our work. We provide detailed responses below to address your concerns.
>
> ***
>
> > W1. "Novelty. The E-M training between LLMs and GNNs has been explored by a handful of pioneer works, e.g., SimTEG and GLEM. The proposed CurGL enhances such framework by adding additional strategies for balancing the LLM and the GNN."
>
> The previous GraphLLM(e.g. GLEM[11]) paper employs the EM algorithm for jointly training LLM and GNN. However, our motivation lies in addressing the issue of text-structure imbalance in node representation learning. While the EM algorithm serves as a classical approach, our Curriculum Co-play Alignment method introduces the following unique aspects:
>
> 1.	**Curriculum strategies tailored to text-structure imbalance**: To tackle the imbalance issue, we design different curriculum strategies for the GNN and LLM within the EM framework. Specifically, the GNN processes structure-difficulty nodes progressively, from easy to hard, while the LLM focuses on text-difficulty nodes. This ensures both components undergo comprehensive training and avoid shortcuts.
>
> 2.	**Iterative enhancement with confident pseudo-labels**: A key objective is to iteratively integrate valuable information between the LLM and GNN, progressively enriching both components with a balance of textual and structural insights. Unlike methods that rely on random pseudo-label sampling often introducing significant label noise, our curriculum strategy emphasizes the selection of confident pseudo-labels to facilitate mutual learning between the components. As shown in Figures 5(a) and 5(b), our method effectively selects high-confidence pseudo-labels.
>
> ***
>
> > W2. "Unclear and potentially unfair experimental setting for baseline llm-as-predictor models. Is is unclear that how are the llm-as-predictor models testified, especially in Table 1. Is is based on a zero-shot manner as in Glbench, or are they also trained with the same training set? In terms of training, is it training-from-scratch or fine-tuning?"
>
> 1. "Unclear and potentially unfair experimental setting for baseline llm-as-predictor models.Is is unclear that how are the llm-as-predictor models testified, especially in Table 1. Is is based on a zero-shot manner as in Glbench, or are they also trained with the same training set?"
>
>     To ensure a fair comparison, all our baselines strictly follow the supervised learning settings defined in GLBench[7]. Specifically, for the LLM-as-predictor methods, we train them using the same training dataset as the baselines and our method. We replace the LLMs in the LLM-as-predictor methods with Llama2-7B. This is because the different LLMs used in the original papers might influence the results.
>     All models are trained on the same training set as the other baselines. For the LLM training part, we strictly follow the guidelines in the original papers, respecting whether LLM training is included or not.
>
>
> 2. "In terms of training, is it training-from-scratch or fine-tuning?"
>
>     We fine-tune all models using the publicly available parameters of LLaMA2-7B, ensuring consistency and reproducibility across experiments.
>
> ***
>
> >  W3. "Lack of discussion of a very relevant work, GraphGPT: Graph Instruction Tuning for Large Language Models."
>
> Thank you for your suggestion. In response, we have added GraphGPT [1] in Appendix Section B under Related Work. **GraphGPT** leverages instruction tuning to adapt LLMs for downstream graph tasks, utilizing natural language alongside a graph-text aligner to effectively encode and communicate the structural information of graphs. This approach enables LLMs to understand complex graph structures and enhances their adaptability across diverse datasets and tasks. Additionally, we have included several other relevant studies [2–5].

---

> ### Author Response · Authors · 2024-11-24
>
> ***
> > W4. "The proposed CurGL relies heavily on the availability of training data, and shows poor potential in transferring learning and zero-shot generalisation. Although it leverages an LLM for textual-based learning, the design of node-wise difficulty measurer and class-based selection strategy requires high-quality labels for the model to learn and converge. While it is quite reasonable for a graph learning model, the trained CurGL cannot be adapted to unseen data (unlike LLaGA, OFA and others), making its potential application quite narrow."
>
> 1. "transferring learning, zero-shot generalisation and unseen dataunseen data"
>
>     Our framework is designed to enable LLMs and GNNs to effectively learn both textual and structural representations of nodes. Following the GLBench setup, we use node classification as the primary task to demonstrate the effectiveness of our method. In the future, we plan to extend our approach to other tasks.
>     But now, many existing GraphLLM methods leverage LLMs as text encoders (e.g., OFA[6]). Our fine-tuned LLM can be effectively integrated into these methods, enhancing their performance in transfer learning, zero-shot tasks, and generalization to unseen data.
>
>
> 2. "the design of node-wise difficulty measurer and class-based selection strategy requires high-quality labels for the model to learn and converge"
>
>     We have conducted comparative experiments under low-quality label conditions in Appendix section E. The results show that our method performs well even with low-quality labels. This is because our approach samples high-quality pseudo-labels as Figure 5 (a) and (b), thereby mitigating the impact of noise in the training set.
>
> | Model   | Cora  |       |       |       | Citeseer |       |       |       | Pubmed |       |       |       |
> |---------|----------------------|-------|-------|-------|-------------------------|-------|-------|-------|-----------------------|-------|-------|-------|
> |         | 0.2             | 0.3   | 0.4   | 0.5   | 0.2                 | 0.3   | 0.4   | 0.5   | 0.2              | 0.3   | 0.4   | 0.5   |
> | GCN     | 66.82               | 49.75 | 45.59 | 50.58 | 54.87                  | 48.63 | 46.14 | 51.16 | 64.77                | 47.30 | 54.12 | 53.85 |
> | GAT     | 66.68               | 57.49 | 49.32 | 47.82 | 58.18                  | 47.15 | 49.61 | 41.62 | 64.44                | 50.22 | 51.79 | 49.89 |
> | SAGE    | 65.81               | 58.17 | 50.62 | 49.41 | 56.70                  | 54.71 | 45.01 | 38.46 | 61.83                | 61.77 | 47.40 | 48.05 |
> | RCL     | 73.01               | 64.50 | 55.46 | 59.13 | 55.41                  | 52.61 | 48.32 | 44.34 | 74.23                | 66.41 | 55.34 | 49.04 |
> | TSS     | 75.13               | 68.11 | 59.20 | 61.92 | 58.12                  | 49.62 | 48.69 | 45.20 | 71.66                | 56.43 | 54.12 | 53.51 |
> | GLEM    | 77.56               | 76.01 | 75.19 | 72.53 | 64.84                  | 63.36 | 60.01 | 54.79 | 82.17                | 78.90 | 75.50 | 73.61 |
> | CurGL   | 79.40               | 78.33 | 76.30 | 74.51 | 69.60                  | 65.15 | 63.21 | 60.91 | 83.61                | 79.66 | 78.45 | 77.18 |
> ***

---

> ### Author Response · Authors · 2024-11-24
>
> >  W5. "Can CurGL generalise beyond node classification, to link prediction or graph classification?"
>
> Many recent studies (e.g., [7-11]) focus on learning for text-attributed graph node classification. Our method emphasizes jointly learning both textual and structural features of nodes to learn effective node representations, which could be used for a variety of applications such as node classification and link prediction. The text embeddings generated by our method (LLM component) can serve as initial features for GNNs, and our fine-tuned LLM can be transferred to other methods that use LLMs as text encoders (e.g., OFA [6]), making them adaptable to various downstream tasks such as link prediction and graph classification.
>
> ***
>
> We truly hope we’ve answered all your questions. Thank you for your support, and we’re happy to discuss further if there’s anything we missed. Thank you again!
>
> Best regards,
>
> The authors
>
> ***
>
> Reference:
>
> [1] Tang, Jiabin, et al. "Graphgpt: Graph instruction tuning for large language models." Proceedings of the 47th International ACM SIGIR Conference on Research and Development in Information Retrieval. 2024.
>
> [2] Huang, Chao, et al. "Large Language Models for Graphs: Progresses and Directions." Companion Proceedings of the ACM on Web Conference 2024. 2024.
>
> [3] Tang, Jiabin, et al. "Higpt: Heterogeneous graph language model." Proceedings of the 30th ACM SIGKDD Conference on Knowledge Discovery and Data Mining. 2024.
>
> [4] Wei, Wei, et al. "Llmrec: Large language models with graph augmentation for recommendation." Proceedings of the 17th ACM International Conference on Web Search and Data Mining. 2024.
>
> [5] Guo, Zirui, et al. "Graphedit: Large language models for graph structure learning." arXiv preprint arXiv:2402.15183 (2024).
>
> [6] Liu, Hao, et al. "One for all: Towards training one graph model for all classification tasks." ICLR 2024.
>
> [7] Li, Yuhan, et al. "Glbench: A comprehensive benchmark for graph with large language models." NeurIPS 2024.
>
> [8] Chen, Zhikai, et al. "Label-free node classification on graphs with large language models (llms)." arXiv preprint arXiv:2310.04668 (2023).
>
> [9] Zhu, Yun, et al. "Efficient tuning and inference for large language models on textual graphs." CoRR 2024.
>
> [10] He, Xiaoxin, et al. "Harnessing explanations: Llm-to-lm interpreter for enhanced text-attributed graph representation learning." ICLR 2024.
>
> [11] Zhao, Jianan, et al. "Learning on large-scale text-attributed graphs via variational inference." ICLR 2023.

---

> ### Comment · Reviewer_t915 · 2024-11-26
>
> Thanks for the authors' detailed response. I appreciate the efforts the authors have made to clarify on some of the concerns. The response addressed most of my questions. However, based on the description that "We replace the LLMs in the LLM-as-predictor methods with Llama2-7B. We fine-tune all models using the publicly available parameters of LLaMA2-7B, ensuring consistency and reproducibility across experiments.", I still think such a setting should be reconsidered. Directly re-initialize the parameters with Llama2-7B and train these Graph LLMs from scratch seems to diminish the advantage of large-scale graph-text SFT as a pre-training staged considered for Graph LLMs, and which is also fundamental for Graph LLMs. We need to reconsider whether further fine-tuning on the released checkpoints of these models would serve as a better comparison method. I understand under the current settings, it is convincing that CurGL derives more expressive embeddings for graph tasks with exactly the same amount of data. In that case, plugging CurGL to SOTA Graph LLMs as featurizers (like CLIP in MLLMs) seems to be more reasonable and easier to scale up. In all, I'll keep my vote for the current version, and I truly feel that this version has much space for improvement.

---

> > ### Author Response · Authors · 2024-12-02
> > **Kindly Reminder**
> >
> > Dear Reviewer t915,
> >
> > I hope this message finds you well. I wanted to kindly remind you that the discussion period for our ICLR submission titled “Curriculum GNN-LLM Alignment for Text-Attributed Graphs” is approaching its conclusion.
> >
> > We truly value your insights and feedback, and your response would greatly contribute to refining and strengthening our work. If there are any remaining questions or clarifications we can address to assist with your feedback, please don’t hesitate to reach out.
> >
> > We understand how busy this time can be and sincerely appreciate the effort you invest in the review process. We look forward to your input and hope to hear from you soon.
> >
> > Best regards,
> > The authors

---

> ### Author Response · Authors · 2024-11-30
>
> Dear Reviewer **t915**
>
> ***
>
> Thank you very much for your valuable feedback, which has deepened my understanding of Graph LLMs. I fully agree that “the advantage of large-scale graph-text SFT as a pre-training staged considered for Graph LLMs”.  However, I believe this represents only one part of the broader scope of Graph LLMs. In our paper, we follow the definition of GraphLLM as outlined in GLBench[1], where GraphLLM are defined as models that integrate LLMs and graphs to learn TAGs.
>
> I believe current Graph LLM research can be broadly categorized into two parts. The first focuses on using LLMs for large-scale graph-text SFT, achieving strong performance across various domains/tasks, as you have aptly pointed out. The second focuses on processing graph-text data by leveraging LLMs/LMs to understand the textual modality rather than emphasizing their cross-domain and cross-task applications after large-scale pretraining. Both directions undoubtedly have their merits. Our work emphasizes the latter, where LLMs/LMs serve primarily as encoders for processing textual modality.
>
> I deeply appreciate your suggestion, which has inspired us to explore whether graph-text SFT as a pre-training stage could enhance the performance of “LLM as a predictor,” potentially surpassing the “LLM as an enhancer” approach trained on a single dataset. Notably, we observe that some “LLM as predictor” baselines do not fine-tune their LLMs. Therefore, we select LLAGA[2] as a representative work that involves fine-tuning. Moreover, due to differences in the training/testing splits between the released pre-training parameters and our setup, there is a possibility that our test set overlaps with the training set used in the original paper. To address this, we pretrain LLAGA on our datasets and finetune it on individual datasets. The results are shown as below, indicating that LLAGA still falls significantly short of the performance achieved in our experiments. Due to time constraints, we are unable to conduct a more extensive comparison. We will further explore this in future work.
>
> | Model  | Cora Acc  | Citeseer Acc |  Pubmed Acc |
> |--------|----------|---------|--------------|
> | CurGL  | 85.49     | 73.92         | 85.07      |
> | LLAGA  | 75.77     | 59.08           | 58.52      |
>
> We sincerely appreciate your recognition of our work and your thoughtful suggestions. Your proposal to integrate **CurGL** into SOTA Graph LLMs as featurizers is exceptionally insightful. However, the current lack of a unified framework for Graph LLMs presents significant challenges in implementing it. We are eager to explore this direction further in our future work.
>
> ***
> [1] Zhao, Jianan, et al. "Learning on large-scale text-attributed graphs via variational inference." ICLR 2023.
>
> [2] Chen, Runjin, et al. "Llaga: Large language and graph assistant." ICML 2024.
> ***
> Best regards,
>
> The authors

---

### Comment · Area_Chair_L3Qt · 2024-11-25
**Please reply to the authors' response.**

Dear reviewers,

The ICLR author discussion phase is ending soon. Could you please review the authors' responses and take the necessary actions? Feel free to ask additional questions during the discussion. If the authors address your concerns, kindly acknowledge their response and update your assessment as appropriate.


Best,
AC

---

### Author Response · Authors · 2024-12-04
**General Response to All Reviewers(1/2)**

We sincerely appreciate all reviewers' insightful and valuable suggestions in reviewing our paper.    We are truly grateful for your recognition of our contributions:

| **Aspect**                     | **Comment**                                                  | **Reviewer**  |
| ------------------------------ | ------------------------------------------------------------ | ------------- |
| **Motivation**                 | "The motivation to balance the role of LLMs and GNNs in a fine-grained sample-wise manner for TAG learning is straightforward and convincing." | Reviewer t915 |
|                                | "The integration of node difficulty into node classification is clearly motivated and engaging." | Reviewer S9p5 |
| **Presentation**               | "Writing is clear and easy to follow. Pseudo code is present to assist better understanding of the mechanisms designed." | Reviewer t915 |
|                                | "The presentation is well-organized, and the ablation study clearly demonstrates the impact of each technique." | Reviewer S9p5 |
|                                | "The paper is clearly written and easy to follow."           | Reviewer ZPUW |
|                                | "The writing is fluent and easily comprehensible."           | Reviewer s8oL |
| **Novelty**                    | "It is one of the earliest works to apply Curriculum Learning in the field of TAGs." | Reviewer s8oL |
|                                | "The authors propose a class-based hard node selection strategy and a text-structure difficulty measurer, which serve as an inspiration for the community." | Reviewer s8oL |
| **Experimental Effectiveness** | "The proposed method demonstrates improved performance."     | Reviewer ZPUW |

***

The reviewers also raised some concerns about our work. Here, we aim to address a few common concerns and will provide detailed responses to others individually.

> R1: The differences between the **EM training** in this paper and previous works.

The previous GraphLLM(e.g. GLEM[1]) paper employs the EM algorithm for jointly training LLM and GNN. However, our motivation lies in addressing the issue of text-structure imbalance in node representation learning. While the EM algorithm serves as a classical approach, our Curriculum Co-play Alignment method introduces the following unique aspects:

1. **Curriculum strategies tailored to text-structure imbalance**: To tackle the imbalance issue, we design different curriculum strategies for the GNN and LLM within the EM framework. Specifically, the GNN processes structure-difficulty nodes progressively, from easy to hard, while the LLM focuses on text-difficulty nodes. This ensures both components undergo comprehensive training and avoid shortcuts.
2. **Iterative enhancement with confident pseudo-labels**: A key objective is to iteratively integrate valuable information between the LLM and GNN, progressively enriching both components with a balance of textual and structural insights. Unlike methods that rely on random pseudo-label sampling often introducing significant label noise, our curriculum strategy emphasizes the selection of confident pseudo-labels to facilitate mutual learning between the components. As shown in Figures 5(a) and 5(b), our method effectively selects high-confidence pseudo-labels.

---

> ### Author Response · Authors · 2024-12-04
> **General Response to All Reviewers(2/2)**
>
> > R2: The Experiemental settings.
>
>
>
> **Baseline settings**
>
> To ensure a fair comparison, we follow GLbench[2] (published in NeurIPS 2024), a widely recognized benchmark for GraphLLM to standardize the LLMs and GNNs used in both GraphLLM baselines and our method. Specifically, in GLbench, the GNNs in the GraphLLM baselines are replaced with GraphSAGE, the LLMs in the LLM-as-predictor methods are replaced with LLaMA2-7B, the LLMs used as text encoders in the GraphLLM baselines are replaced with Sentence-BERT, and BERT is employed to process raw text into embeddings for Vanilla GNNs (e.g., GCN, GAT, and GraphSAGE), which is also adopted in our method. Since we share the same goal of ensuring a fair comparison as GLbench, we follow their setting.
>
> **GCN outperforms some GraphLLM methods, especially in Cora dataset.**
>
> Reviewing our table, GCN consistently outperforms GraphSAGE by 1–3% across all datasets. To ensure a fair comparison of GraphLLM methods, we replaced all GNNs in the GraphLLM baselines with GraphSAGE. This might explain why GCN performs particularly well on the **Cora** dataset, surpassing almost all GraphLLM baselines. However, it is evident that GCN still falls short compared to the best GraphLLM baselines on other datasets. Another possible reason for GCN’s relatively strong performance is that the initial textual input of the **Cora** dataset is already well-understood. As a result, adding LLM/LM enhancements may not yield significant improvements. Our experimental results are consistent with observations in GLbench[2].
>
> **More Experiements beyond node classification.**
>
> Our purpose is to learn a text-structure balance node representation, as node classification is  the most relevant task for evaluating node representations. we have followed a well-known GraphLLM benchmark GLBench[2] that uses node classification as the evaluation task, We believe that our method’s performance surpassing this benchmark have effectively demonstrated its capability in learning node representations, aligning well with our purpose. Due to the limited time available during the rebuttal period, we may not be able to provide results for other tasks in a timely manner, as the GraphLLM baseline requires significant computational resources. We plan to extend our method beyond node classification in future work.
>
> ***
>
> Compared to the initial version, we have made the following improvements based on the reviewers’ suggestions:
>
> 1. Thanks to **Reviewer t915**, We have added comparative experiments under low-quality label conditions in Appendix E. The results show that our method performs well even with low-quality labels. This can be attributed to our approach samples high-quality pseudo-labels as Figure 5 (a) and (b), thereby mitigating the impact of noise in the training set.
> 2. We have added the time efficiency of our method with the baselines and included additional experimental analyses in the Appendix F. The experimental results show that our method achieves a significant improvement over the baselines, with an acceptable time consumption.
> 3. Thank you to **Reviewer ZPUW** for the suggestion to present real experimental results to support the text-structure imbalance problem. We have added an experiment in Appendix G to illustrate the issue of text-structure imbalance. Specifically, we trained the LLM and GNN separately and evaluated their performance on the all nodes. The results show that some nodes are correctly predicted by the LLM but incorrectly predicted by the GNN, while others are correctly predicted by the GNN but incorrectly predicted by the LLM. This demonstrates the existence of the text-structure imbalance problem.
> 4. Thanks to **Reviewer s8oL** for the suggestion. We have included detailed explanations and mathematical proofs related to the EM algorithm in Appendix H.
>
>
>
> ***
>
> We sincerely thank the reviewers for their valuable suggestions, which have strengthened our work. We hope our responses address any confusion and help alleviate your concerns.
>
> Best regards,
>
> The authors
>
> ***
>
> Reference:
>
> [1] Zhao, Jianan, et al. "Learning on large-scale text-attributed graphs via variational inference." ICLR 2023.
>
> [2] Li, Yuhan, et al. "Glbench: A comprehensive benchmark for graph with large language models." NeurIPS 2024.

---

### Meta-Review · Area_Chair_L3Qt · 2024-12-13

**Metareview:**

The paper addresses the text-structure imbalance problem in Text-Attributed Graphs (TAGs) by aligning Graph Neural Networks (GNNs) and Large Language Models (LLMs) using a curriculum learning approach. The proposed Curriculum GNN-LLM Alignment (CurGL) method balances the learning difficulties of textual and structural information on a node-by-node basis, enhancing the alignment between GNNs and LLMs and improving performance on node classification tasks.

The review highlighted the paper’s strengths, including clear writing, an innovative approach, and improved performance. However, concerns were raised about incremental novelty, computational complexity, and the limited scope of experiments, which focused only on node classification. Reviewers also had questions about the selection of hyperparameters, explanation of trends in results, and computational efficiency. The rebuttal seems to partially address the concerns. Most reviewers did not change their ratings. Overall comments suggest a weak rejection.

**Additional Comments On Reviewer Discussion:**

The rebuttal seems to partially address the concerns. Unfortunately, most reviewers did not change their ratings. Only one reviewer raised the score. Overall comments suggest a weak rejection.

---

### Decision · Program_Chairs · 2025-01-22

Reject